**Technology**

# Atlas-scale single-cell DNA methylation profiling with sciMETv3

## Graphical abstract

## Authors

Ruth V. Nichols, Lauren E. Rylaarsdam,
Brendan L. O'Connell, Zohar Shipony,
Nika Iremadze, Sonia N. Acharya,
Andrew C. Adey

## Correspondence

adey@ohsu.edu

## In brief

DNA methylation is a foundational layer of epigenetic regulatory control. Nichols and Rylaarsdam et al. developed a technology capable of producing hundreds of thousands of single-cell DNA methylation profiles in a single experiment. The technology is versatile to multiple sequencing platforms, and a variant allows for the profiling of both DNA methylation and the accessible chromatin landscape from the same cells.

## Highlights

- Atlas-scale production of single-cell DNA methylation in a single experiment

- Evaluation using Illumina and Ultima Genomics sequencing platforms

- Integrated >140,000 cell dataset of human cortex from four individuals

- Chromatin accessibility and genome-wide DNA methylation from the same cells

 Nichols et al., 2025, Cell Genomics 5, 100726
January 8, 2025 © 2024 The Author(s). Published by Elsevier Inc.

CellPress

## Technology

# Atlas-scale single-cell DNA methylation profiling with sciMETv3

Ruth V. Nichols,[1,6] Lauren E. Rylaarsdam,[1,6] Brendan L. O'Connell,[1,2] Zohar Shipony,[3] Nika Iremadze,[3] Sonia N. Acharya,[1] and Andrew C. Adey[1,2,4,5,7,*]

[1]Department of Molecular & Medical Genetics, Oregon Health & Science University, Portland, OR, USA
[2]Cancer Early Detection Advanced Research Institute, Oregon Health & Science University, Portland, OR, USA
[3]Ultima Genomics, Fremont, CA, USA
[4]Knight Cardiovascular Institute, Oregon Health & Science University, Portland, OR, USA
[5]Knight Cancer Institute, Oregon Health & Science University, Portland, OR, USA
[6]These authors contributed equally
[7]Lead contact
*Correspondence: adey@ohsu.edu

## SUMMARY

Single-cell methods to assess DNA methylation have not achieved the same level of cell throughput per experiment compared to other modalities, with large-scale datasets requiring extensive automation, time, and other resources. Here, we describe sciMETv3, a combinatorial indexing-based technique that enables atlas-scale libraries to be produced in a single experiment. To reduce the sequencing burden, we demonstrate the compatibility of sciMETv3 with capture techniques to enrich regulatory regions, as well as the ability to leverage enzymatic conversion, which can yield higher library diversity. We showcase the throughput of sciMETv3 by producing a >140,000 cell library from human middle frontal gyrus split across four multiplexed individuals using both Illumina and Ultima sequencing instrumentation. Finally, we introduce sciMET+ATAC to enable high-throughput exploration of the interplay between chromatin accessibility and DNA methylation within the same cell.

## INTRODUCTION

Mammalian DNA methylation takes the form of a methyl group covalently added to the 5 carbon of cytosine residues in the genome and forms the most basal layer of gene regulatory control, with distinct programs that shape the permissible genomic landscape during development. Historically, DNA methylation has been profiled using "conversion"-based approaches, which leverage chemical or enzymatic processes to convert non-methylated cytosines to uracil. Converted bases are then sequenced as thymine, whereas methylated cytosines are protected from this process. The complexity of conversion protocols makes single-cell approaches particularly challenging, with most methods requiring the deposition and processing of individual cells into their own reaction compartments for conversion and then initial processing steps.[1–5] Large-scale efforts, such as those carried out by the NIH BRAIN Initiative, have achieved atlas-scale datasets[6–9]; however, these studies required extensive automation, time, and financial resources to process the thousands of microwell plates and column cleanups, all of which are well beyond what a typical lab could accomplish. The profound insight into neuronal DNA methylation biology and value as a reference atlas of these efforts motivate the need for a technology that can be deployed by diverse research programs and enable the extension of such insight into other areas of biology. We previ-

ously developed techniques to increase the cell throughput for profiling DNA methylation, sciMET[10] and sciMETv2,[11] which leverage single-cell combinatorial indexing to pre-index cells prior to conversion and the final stages of library preparation. This workflow enables the production of thousands of single-cell methylation libraries to be produced by a single individual and distributes reagent costs over many pre-indexed cells, substantially reducing costs per cell. We also demonstrated the ability to perform target capture on regulatory loci with high levels of expected cell-type-specific methylation variability (sciMET-cap), which reduces the number of sequencing reads required per cell to achieve cell type identification and robust cell type clustering.[12]

The sciMETv2 technology can achieve a modest scale of throughput, with typical experiments producing between 5,000 and 20,000 single-cell profiles. This capacity is suitable for many applications; however, to achieve the higher end of that range, multiple plates of indexed tagmentation must be performed, which can be cumbersome and expensive. Here, we directly address these remaining challenges by developing sciMETv3, which leverages an additional tier of cell barcoding to increase throughput by orders of magnitude. The final cell count is flexible and spans three orders of magnitude from ~1,000 to up to 10 million in increments of ~1,000 cells. This technology requires comparable hands-on time in sciMETv2 and produces

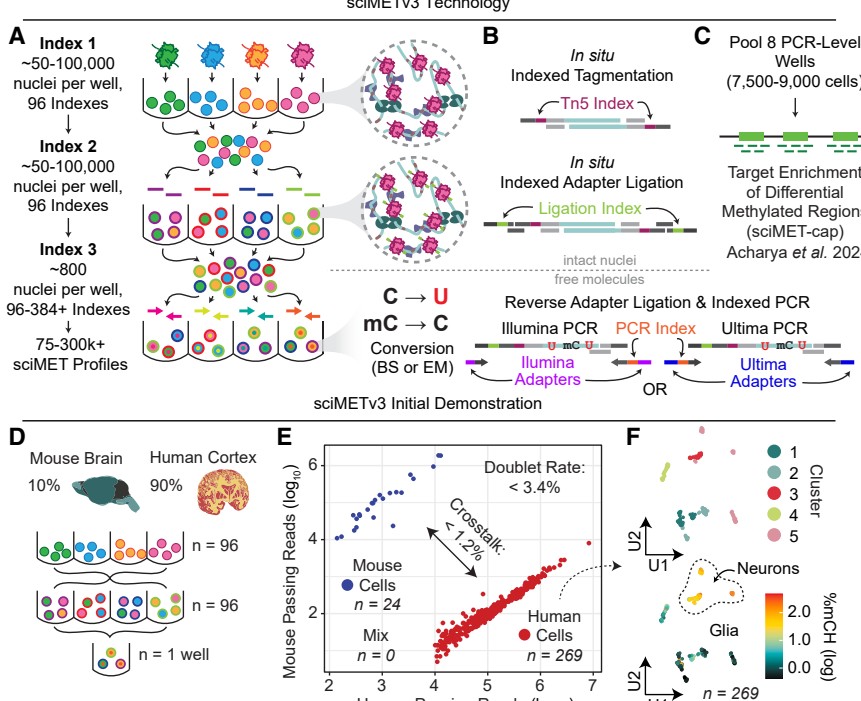

**Figure 1. sciMETv3 technology development**

(A) Indexing and pooling schematic for sciMETv3.

(B) Molecular schematic.

(C) Strategy for sciMET-cap enrichment strategy.

(D) Experimental design schematic for initial sci-METv3 development.

(E) Assessment of doublet rates and cell-cell crosstalk from human and mouse cells.

(F) Uniform manifold approximation and projection (UMAP) of human cells from the initial experiment reveals clear clusters (top) with expected mCH patterns for glia and neurons (bottom).

increasing the throughput of the sci-METv2 platform by 96-fold. The workflow was carried out on four human brain specimens (cortex, BA 46; 90% of nuclei) and a mouse brain specimen (whole brain, C57BL/6; 10% of nuclei), allowing us to estimate our cell doublet rate while providing enough human cells for an initial analysis (Figure 1D). We then processed a single final PCR well out of a total of 8 that were diluted, which produced 293 passing cell profiles with a mean unique read count of 354,763, a mean coverage of 2.73 million total cytosines covered per cell, and a minimal strand bias (49.6% of reads aligned to top strand). Of these, 269 were human and 24 were mouse, with zero cells identified as doublets, establishing a maximum doublet bound of 3.4% when factoring in the 10-fold skewing toward human cells (Figure 1E). We next assessed the crosstalk by measuring the percentage of cross-species aligned reads, also adjusting for the skewed species mixture, resulting in a maximum of 1.2%. Human cells were taken through windowing and clustering. Leveraging both methylation levels of cytosines in the CG-context (mCG) and CH-context (mCH) produced two neuronal and three glial clusters, which were annotated based on global CH methylation levels (Figure 1F).

an identical molecular structure, allowing for capture techniques to be carried out. We further demonstrate the ability to perform enzymatic conversion, as well as a modified workflow to enable libraries to be sequenced on the Ultima Genomics platform. We then combine datasets sequenced by both platforms to produce >140,000 cells from middle frontal gyrus across four healthy human donors. Finally, we demonstrate a variant of the technology that employs two rounds of indexed tagmentation followed by sciMETv3 processing to capture chromatin accessibility via the Assay for Transposase Accessible Chromatin (ATAC) plus genome-wide DNA methylation profiles from the same cells in high throughput.

## DESIGN

To achieve increased throughput for the sciMET platform, we devised a strategy to incorporate an additional round of indexing post-tagmentation and prior to distribution into the final PCR-indexed wells (Figure 1A). This approach was based on a ligation workflow similar to that achieved for sci-ATAC-seq3.[13] Ligation adapters were designed to directly append to the transposase adapter sequence, completing the 5′ half of the Illumina read 2 sequencing primer. These adapters also append a well-specific barcode and terminate with the Illumina flow cell primer sequence at the 5′ end. The final ligation product results in the same final molecular structure that is produced during PCR for the sciMETv2 workflow, retaining compatibility with downstream capture methods (Figures 1B and 1C). The ligation adapters must survive bisulfite conversion and were therefore fully methylated at all cytosine positions. As an initial assessment, we leveraged a set of 96 indexed primers, effectively

## RESULTS

### sciMETv3 is compatible with enzymatic conversion methods as well as target capture

We next assessed the full workflow and platform versatility of sciMETv3 by carrying out a preparation on a human brain specimen (cortex, BA 46) with the goal of expanding into enzymatic-based conversion as well as leveraging our previously described sciMET-cap technology that enriches for a roughly 125 Mbp set of regulatory loci in order to enrich for higher variable regions and subsequently reduce the sequencing depth required to achieve comparable cluster granularity.[12] We leveraged 96 tagmentation and ligation indexes and distributed a target of 750 pre-indexed nuclei into each well of a final plate (Figure 2A). Unlike sciMETv2, the greater number of nuclei within each final well allows for dilution to be deployed as opposed to flow sorting, reducing the overall time of the experiment and eliminating the

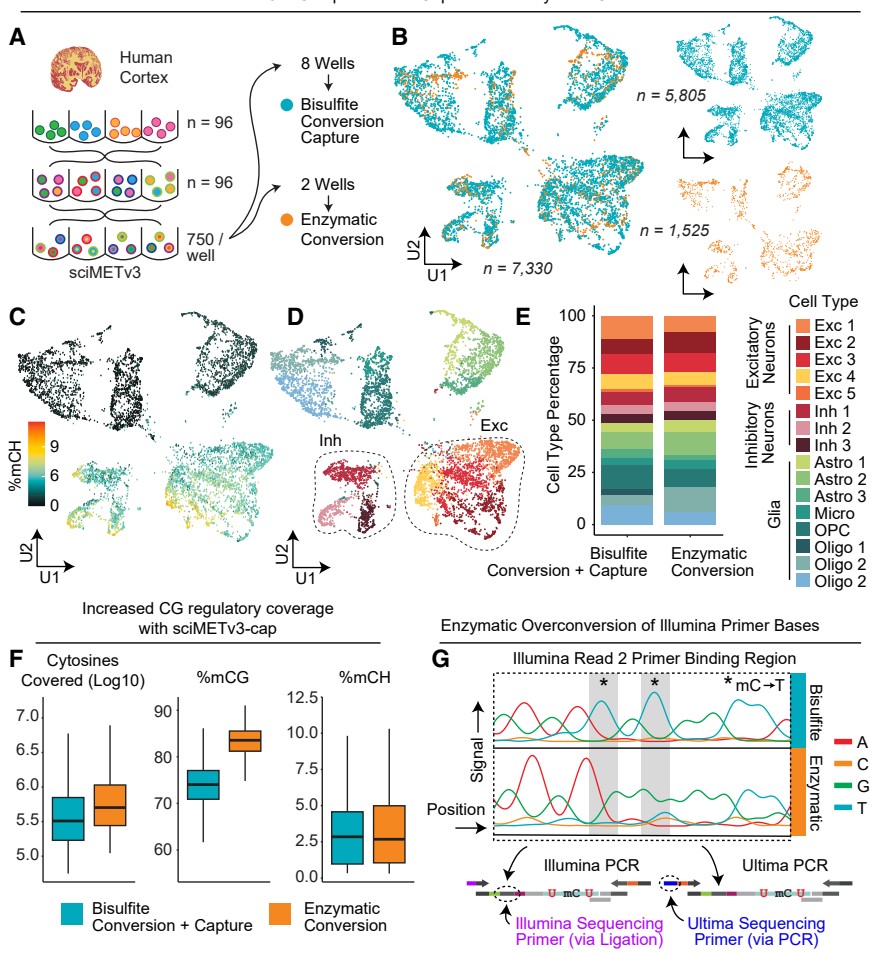

sciMETv3 is Compatible with Capture and Enzymatic Conversion

Increased CG regulatory coverage
with sciMETv3-cap

Enzymatic Overconversion of Illumina Primer Bases

**Figure 2. sciMETv3 is compatible with capture methods and enzymatic conversion**

(A) Experimental design schematic.

(B) UMAP of cells combined from both sciMET-cap using bisulfite conversion and non-captured enzymatic conversion preparations.

(C) mCH levels show expected patterns for neurons and glial cell populations.

(D) Identified clusters with inhibitory and excitatory neuron clusters highlighted.

(E) Cluster proportions are comparable between bisulfite + capture and enzymatic non-captured conditions.

(F) Global methylation patterns show the expected trend, with cells taken through capture exhibiting lower mCG levels due to the enrichment at regulatory loci with no impact on mCH levels. Box and whiskers represent median and quartiles at each increment (box contains 50%, whiskers contain 75%).

(G) Sanger sequencing traces of enzymatic converted libraries show over-conversion of key bases present in the read 2/index read 1 Illumina sequencing primer region that is appended during adapter ligation.

1,525 QC-passing single-cell methylomes. As anticipated, the insert size of library fragments from the enzymatic conversion library was greater than that of bisulfite methods (mean = 163 ± 121 versus 78 ± 69 bp for enzymatic and bisulfite, respectively; 2.1-fold increase).

We next aggregated cell profiles from both the bisulfite-converted sciMETv3-cap experiment and the non-capture enzymatic conversion dataset without deploying any bias correction methodologies, producing comparable results for the distribution of cells in a reduced dimension representation, CH methylation distribution, and cell type composition between the experiments (Figures 2B–2E). Consistent with our previous sciMET-cap datasets, CG methylation was reduced compared to the genome-wide dataset due to the enrichment of regulatory regions that frequently exhibit hypomethylation and not due to conversion biases, which showed comparable global CH methylation levels (Figure 2F).

Despite the increased fragment size using enzymatic conversion methods, we noticed a decrease in sequencing run quality with fewer clusters passing filter (<50% versus >90% typically). We suspected that this may be due to the unintentional conversion of sequencing adapter bases for the read 2/index read 1 primer site that lies on the ligation junction between the indexed tagmentation oligo and the indexed ligation oligo. To evaluate this, we performed Sanger sequencing using outer primers that are appended via PCR and not subjected to conversion. This revealed distinct cytosine conversion to uracil at adapter bases present within the read 2 sequencing primer region (Figure 2G). This is likely due to the sequence specificity of the TET2 catalytic domain, which biases its ability to protect

need for flow cytometry instrumentation. Dilution has been developed for a commercialized version of a combinatorial indexing-based single-cell methylation workflow; however, the increased nuclei count of sciMETv3 provides greater robustness at this stage. Eight wells were taken through bisulfite conversion, reverse adapter ligation, and PCR. All eight wells (estimated cell n = 6,000) were taken through the capture workflow followed by sequencing, producing 5,805 quality control (QC)-passing single-cell DNA methylation profiles with comparable target fold enrichment to sciMET-cap (6.2-fold versus 7- to 10-fold[12]), which translates to a similar rate of on-target reads at 27.7% versus 30%–45%.

The increased nuclei count per conversion well for sciMETv3 over sciMETv2 (96-fold greater) brings the total input within the recommended range for enzymatic conversion methods without the need for ultra-low-input modifications. Enzymatic conversion methods have been shown to offer improved yields over the harsh chemical processes of bisulfite conversion[14] and have been demonstrated previously in the context of other sciMET-like protocols.[15] Two wells of the final plate (estimated cell n = 1,500) were taken through enzymatic conversion followed by reverse adapter incorporation and PCR. Sequencing produced

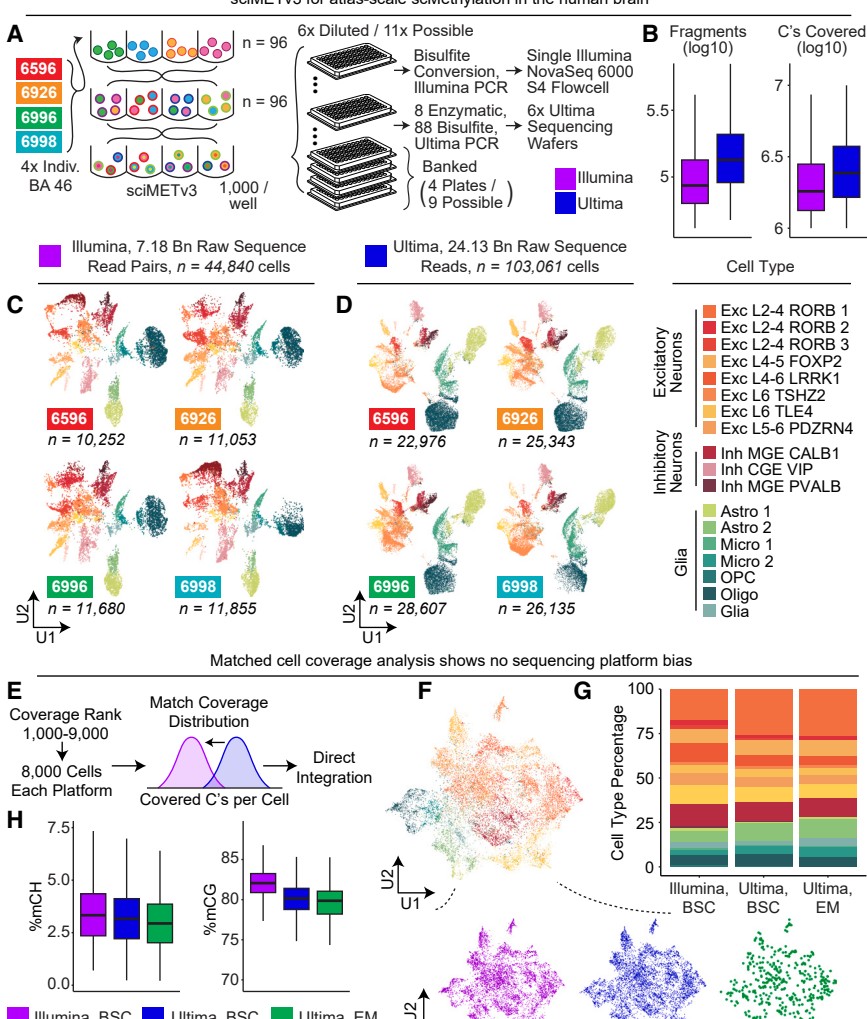

Figure 3. sciMETv3 can produce atlas-scale datasets using Illumina or Ultima sequencing platforms

(A) Experimental design schematic.
(B) Summary of sequencing depth for each platform. Box and whiskers represent median and quartiles.
(C) UMAP of Illumina-sequenced cells split by four individuals and colored by cell type.
(D) UMAP of Ultima-sequenced cells.
(E) Strategy for matching cell coverage distribution between Illumina- and Ultima-sequenced cells for a direct comparison.
(F) UMAP of integrated coverage-matched cells from both platforms colored by cell type and split by platform (below).
(G) Comparable cell type proportions were achieved for each platform.
(H) Comparable global methylation statistics between platforms. Box and whiskers represent median and quartiles.

methylated cytosines from conversion.[16] A possible solution to this problem would be the use of 5-hydroxymethylcytosine (5hmC, or other chemical modifications) in the adapter oligos to ensure protection; however, such modifications are costly and difficult to synthesize. Alternatively, the use of sequencing instruments that do not leverage this region for sequence read priming would eliminate the issue, such as a design compatible with the Ultima Genomics UG100 instrument.

## Atlas-scale dataset production is possible with sciMETv3 on multiple sequencing platforms

To demonstrate the atlas-scale potential of sciMETv3, we performed a single preparation on human brain specimens of four individuals (cortex, BA 46; 6596, 6926, 6996, and 6998), which were distributed across equal numbers of tagmentation indexes (n = 24 each), providing the sample index in addition to the first tier of cell barcoding. After pooling, splitting and adapter ligation, and then pooling again, we obtained enough nuclei to dilute into 11 full 96-well plates at a target dilution count of 1,000 per well for an estimated potential cell count of just over 1 million. In total, we

diluted nuclei into six plates, four of which were banked for possible future processing (Figure 3A). One plate was carried through bisulfite conversion, adapter ligation, and PCR using primers established in previous experiments that append Illumina sequencing primers. The second plate was processed for Ultima-based sequencing using bisulfite conversion for 88 wells, and 8 wells carried through enzymatic conversion. Adapter ligation was then performed followed by PCR using primers that append sequencing primers specific to the Ultima sequencing platform. Importantly, the Ultima-based PCR (Figure 1B, bottom) is direct Ultima library preparation and not

a platform-conversion approach. Beyond alternate primer sequences, the other major design difference was to append the PCR index on the same side of the molecule as the tagmentation and ligation indexes so that the single-end reads produced by Ultima sequencing will read through all three indexes prior to the genomic DNA insert, maximizing the number of reads that will contain all three index sequences.

The first plate was sequenced on a single S4 flow cell of an Illumina NovaSeq 6000 instrument using a paired 200 cycle kit, producing 7.18 billion raw read pairs after demultiplexing (78.0% yield) and an alignment rate of 94.3%. This resulted in 44,840 total cells called with a median of 1.82 million cytosines covered per cell at a median read duplicate rate of 13.98%, indicating that additional sequencing would yield greater coverage before reaching diminishing returns and increasing the total cell number with more cells reaching minimum coverage thresholds (Figure 3B). Cells were split evenly across the four individuals (mean = 11,210 ± 6.4%), and clustering produced distinct primary cell types that were present in all individuals, consistent with previous observations that cell-type-specific methylation is

the predominant signal that drives dimensionality reduction and clustering in brain single-cell DNA methylation datasets[7,11] (Figures 3C and S1). The additional coverage in these datasets compared to the initial pilot studies allowed us to assess allelic bias that may be present in the sciMETv3 assay. We evaluated called variant positions that are not impacted by bisulfite conversion (A-to-T and T-to-A transversions) and observed a mean non-reference allele frequency of 50.01% ± 0.04%, suggesting minimal allele bias from the assay or alignment.

The plate sequenced using the Ultima Genomics UG100 instrument was processed over six wafers, yielding a total of 28.5 billion raw reads, 24.13 billion after demultiplexing and trimming (included in the demultiplexing process; see STAR Methods) with an alignment rate of 93.7%. The increased read counts over the Illumina-sequenced plate resulted in an increased median number of cytosines covered per cell, 2.58 million, with a commensurate increase in read duplicate rate, 31.49%, producing 103,061 called cells (Figure 3B). Similarly, cells were distributed evenly across all four individuals (mean = 25,765 ± 9.0%), with clustering driven by cell type over inter-individual variation (Figure 3D). The lack of a need to preserve sequence integrity over the Illumina sequencing primer region using the Ultima platform enabled us to process a subset of the final indexing plate ($n = 8$ wells) using enzymatic conversion, which produced comparable coverage and methylation statistics when compared to the bisulfite-converted cells (Figure S2).

To evaluate any potential biases driven by the sequencing platform, we took the highest-covered 9,000 cells and then excluded the top 1,000 from each dataset, resulting in 8,000 cells for each platform. We then downsampled reads from the Ultima Genomics cells to achieve a matched distribution of cytosines covered per cell between each set (Figure 3E). We then directly integrated the datasets without any batch correction methods, taking cells through windowing, dimensionality reduction, and clustering, producing concordant distributions of cells across cell types for each platform, including enzymatic-converted cells (Figures 3F and 3G). We next assessed global methylation levels, which produced comparable CH methylation across both platforms and conversion methods, and a slightly reduced CG methylation level for both Ultima-sequenced conditions compared to the Illumina-sequenced cells (Figure 3H). Taken together, the sequencing platform and conversion method do not appear to produce any significant bias in the datasets.

### Integrated map of single-cell DNA methylation in the middle frontal gyrus from four individuals

We next leveraged all cells across both sequencing platforms to produce an integrated atlas of single-cell DNA methylation in human middle frontal gyrus across four individuals, leveraging Harmony[17] to account for the coverage differences between the two datasets (Figures 4A and 4B). Clustering was performed followed by cell type assignment by correlating to a pre-existing atlas[7] and assessing mCG patterns over canonical marker genes (Figures 4C and 4D). The integrated atlas along with the aggregated cell-type-specific methylome profiles and all associated metadata are available as a downloadable R object for use as a reference map that enables interaction, visualization, and integration using the Amethyst computational framework.[18]

Cell type proportions were consistent across individuals as well as platforms, with the largest variance in the proportion of oligodendrocytes present (Figure 4E). High-resolution aggregated CG methylation tracks were then generated for each cluster, providing a granular view of CG regulatory status genome wide for each cell type. Similar to other epigenetic properties, such as ATAC-seq, DNA methylation status at promoters is varied across canonical marker genes, with some exhibiting cell-type-specific hypomethylation (e.g., *MAG* in oligodendrocytes) and others fully hypomethylated across all cell types. However, cell-type-specific methylation patterning throughout the gene can be highly variable, with hypomethylation extending beyond the promoter and into the gene body, or in the form of focal dips in methylation throughout the gene (Figure 4F).

To characterize these distinct patterns, we assessed cell type clusters ($n = 31$) genome wide for hypomethylated regions (HMRs; STAR Methods). In total, 155,110 distinct HMRs were identified, with 65,161 (42.0%) unique to a single cluster. A total of 106,424 (68.6%) overlapped ENCODE DNaseI hypersensitivity sites[19] (1.38-fold genomic enrichment, $p < 2.2e{-}16$, Hypergeom.), consistent with the majority of these sites serving a regulatory role. 18,800 (12.1%) HMRs overlapped promoter regions (8.07-fold genomic enrichment, $p < 2.2e{-}16$), with only 1,463 (7.8% of promoter HMRs) unique to a cluster, and a mean of 17.5 clusters exhibiting hypomethylation at HMRs, indicating a propensity for cross-cell-type promoter hypomethylation, regardless of expression status. In contrast, of the 44,304 (28.6%) of enhancer-overlapping HMRs (1.74-fold genomic enrichment, $p < 2.2e{-}16$), 14,668 (33.1% of enhancer HMRs) were cell type specific, and a mean of 4.7 cell types exhibited hypomethylation at these HMRs, suggesting increased cell type specificity versus promoter elements, consistent with previous enhancer characterization studies[20] (Figure 4G).

### Stepwise indexed tagmentation enables DNA methylation plus ATAC in single cells

Chromatin accessibility has been proven to be valuable property for assessing the regulatory landscape of cells.[13,19–22] Advances in technology platforms to assess chromatin accessibility using transposase-based workflows (ATAC) at the single-cell level have enabled large-scale atlases to be produced across multiple tissue types, providing a valuable reference resource to aid in cell type assignment via marker gene assessment or direct integration. Previously, technologies have been developed to assay nucleosome occupancy (similar to chromatin accessibility) alongside DNA methylation (NOMe) by pre-treating nuclei using a bacterial DNA methyltransferase to artificially methylate cytosines in the GC context at both accessible chromatin and histone linker regions.[23] Assessing methylation levels at these sites, excluding those in the GCG context, allows the ability to determine if a site was accessible by the methyltransferase. These technologies have been extended to the single-cell level[24–26]; however, the lack of enrichment for regulatory loci, as is the case for ATAC-based technologies, make coverage of the 1%–5% of the accessible genome[19] extremely sparse on a per-cell basis, relegating any chromatin analysis to the cell type or cluster level.[27]

We previously described a technology that enables the assessment of chromatin accessibility (via ATAC) alongside

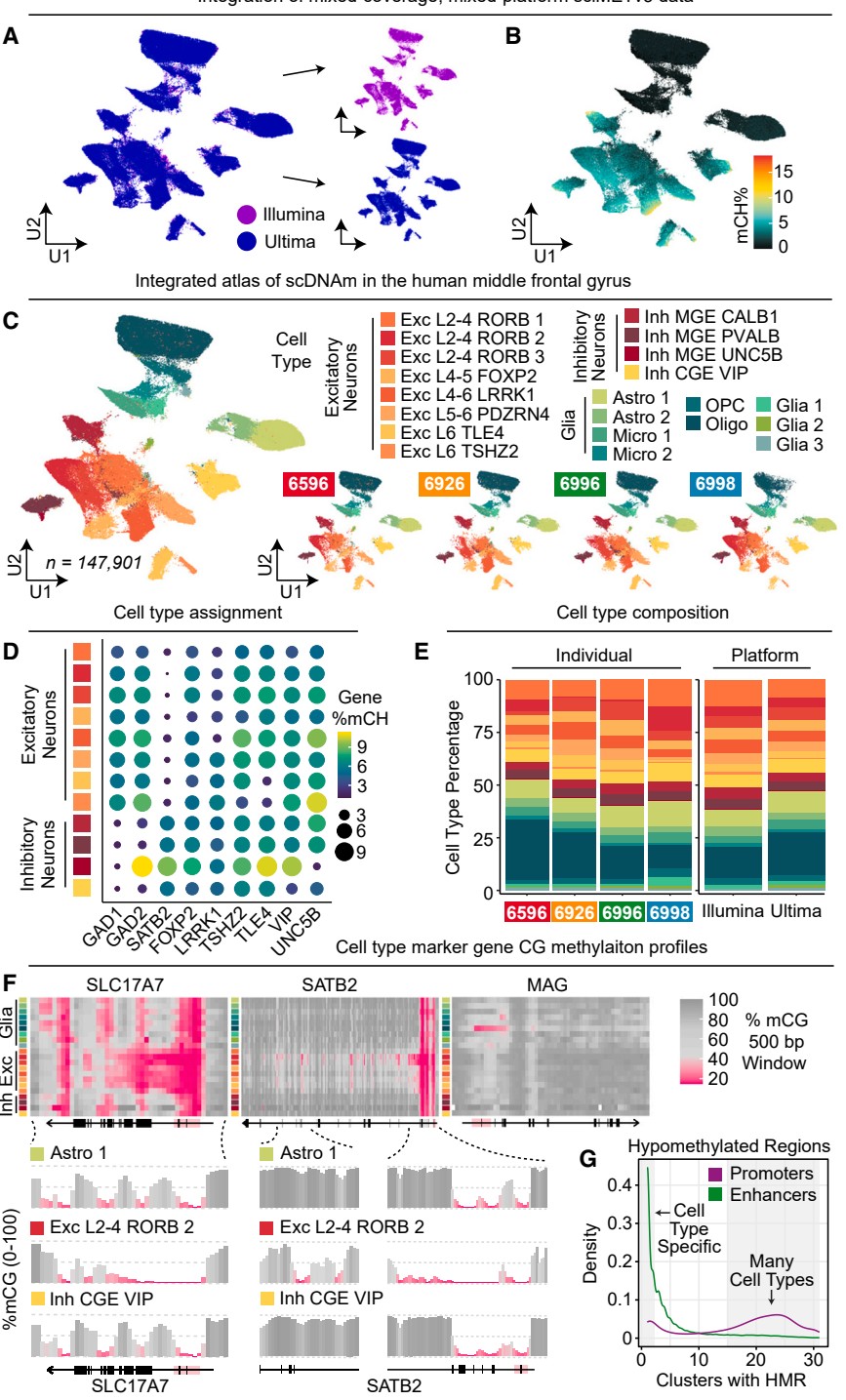

Integration of mixed-coverage, mixed-platform sciMETv3 data

**A** Illumina / Ultima

**B** mCH%

Integrated atlas of scDNAm in the human middle frontal gyrus

**C**

Cell Type

Excitatory Neurons:
Exc L2-4 RORB 1
Exc L2-4 RORB 2
Exc L2-4 RORB 3
Exc L4-5 FOXP2
Exc L4-6 LRRK1
Exc L5-6 PDZRN4
Exc L6 TLE4
Exc L6 TSHZ2

Inhibitory Neurons:
Inh MGE CALB1
Inh MGE PVALB
Inh MGE UNC5b
Inh CGE VIP

Glia:
Astro 1
Astro 2
Micro 1
Micro 2
OPC
Oligo
Glia 1
Glia 2
Glia 3

6596  6926  6996  6998

n = 147,901

Cell type assignment

Cell type composition

**D** Gene %mCH

Excitatory Neurons / Inhibitory Neurons

GAD1 GAD2 SATB2 FOXP2 LRRK1 TSHZ2 TLE4 VIP UNC5B

**E** Cell Type Percentage

Individual: 6596 6926 6996 6998
Platform: Illumina Ultima

Cell type marker gene CG methylaiton profiles

**F** SLC17A7  SATB2  MAG

% mCG 500 bp Window: 100 80 60 40 20

Glia / Inh / Exc

Astro 1 / Exc L2-4 RORB 2 / Inh CGE VIP

%mCG (0-100)

SLC17A7  SATB2

**G** Hypomethylated Regions

Promoters / Enhancers

Cell Type Specific / Many Cell Types

Density / Clusters with HMR

**Figure 4. An atlas of single-cell DNA methylation in human middle frontal gyrus**

(A) Combined UMAP across both sequencing platforms.

(B) Global mCH percentages for the combined dataset.

(C) Combined UMAP colored by cell type and split by individual (right).

(D) Marker gene body mCH levels by cluster.

(E) Cell type proportions across individual and sequencing platforms.

(F) mCG levels across marker genes show distinct cluster-specific patterns.

(G) Enhancers exhibit highly cell-type-specific hypomethylation compared to promoters.

different index. Nuclei were then loaded onto a 10× Genomics Chromium instrument for droplet-based barcoding. Here, we applied a similar concept to our sciMETv3 workflow, performing an initial tagmentation on native human cortex nuclei using one set of 8 indexed sciMET Tn5 complexes. After the first round of tagmentation to encode open chromatin, we then performed fixation, nucleosome disruption, and then a second round of tagmentation using a different set of 8 indexed complexes, which are able to access the rest of the genome. Nuclei were then pooled and taken through the remainder of the sciMETv3 workflow, targeting 90 nuclei for each final well of indexing for an expected 6,480 cell profiles (Figure 5A). Raw sequence reads were demultiplexed using the three tiers of indexing, splitting out the paired ATAC and MET indexes from the first round with 4.79% of reads derived from the first (ATAC) tagmentation and the remaining 95.21% from the second (MET) tagmentation, roughly matching the proportion of accessible versus inaccessible chromatin.[29] In total, 5,305 cells met minimum unique passing read counts for both the ATAC and MET paired datasets with read-depth concordance between the modalities (Figure 5B).

ATAC reads were processed using the standard sciMET processing workflow through alignment. As an initial assessment, peaks were called using Macs2,[30] which produced 147,176 peaks from the 56.1 million total fragments, within the expected range for bulk ATAC-seq studies. Of these, 139,769 (95.00%) overlapped with previously identified accessible genomic loci, suggesting that the majority are likely *bona fide* candidate *cis*-regulatory elements.[19] Fragments were then used as input for SnapATAC2[31] for single-cell-level analysis.

whole-genome sequences (WGSs) from the same cells (scATAC+WGS) by leveraging two rounds of indexed tagmentation.[28] The first round of tagmentation is performed on native nuclei, thus capturing the open chromatin landscape. Subsequent fixation and nucleosome disruption enables the second round of tagmentation to be performed on the rest of the genome using a

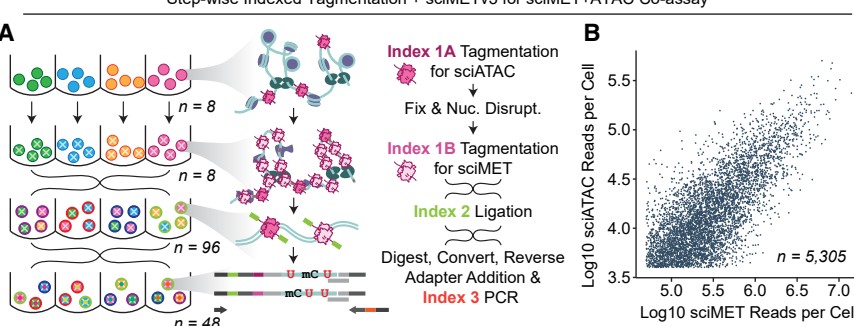

Step-wise Indexed Tagmentation + sciMETv3 for sciMET+ATAC Co-assay

**Figure 5. sciMET+ATAC for joint single-cell DNA methylation and chromatin accessibility**

(A) sciMET+ATAC co-assay schematic.

(B) Concordant ATAC and methylation read counts per cell.

(C) TSSe for the ATAC modality is low yet consistent for the tissue sampled.

(D) UMAP of ATAC modality including a reference atlas and s3-ATAC preparation, split by dataset.

(E) ATAC-based UMAP colored by cell type.

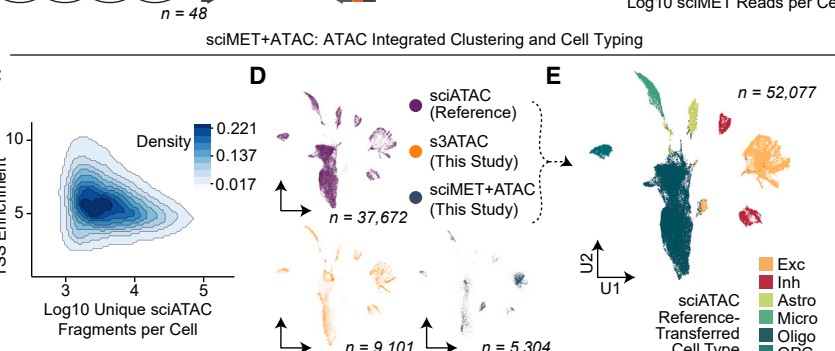

sciMET+ATAC: ATAC Integrated Clustering and Cell Typing

Transcription start site enrichment (TSSe) was relatively low (5.2; Figure 5C) compared to typical single-cell ATAC-seq methods (~10–20)[31]; however, this is expected due to the double-tagmentation nature of the assay. For a typical scATAC workflow, two proximal tagmentation events are required in order to produce a short fragment that can be taken through subsequent library processing, with spurious tagmentation events yielding long fragments that are not able to be amplified in the final PCR stage. In our assay, spurious tagmentation events during ATAC tagmentation are subjected to shortening due to the subsequent genome-wide tagmentation after nucleosome disruption, making them viable for downstream processing.

One valuable utilization of the sciMET+ATAC assay is the ability to leverage the ATAC modality for integration with existing reference atlas datasets where a methylation reference may not be available. We therefore generated an s3-ATAC[32] dataset from the same tissue specimen using a novel implementation of the workflow that utilizes the iCell8 instrument for post-tagmentation processing in a 5,184-nanowell chip, similar to previous workflows for sciATAC.[33] In total, we leveraged a 32 × 32 nanowell setup targeting just under 12 pre-indexed nuclei per well for a total target of 12,000 total s3-ATAC profiles. Sequence reads were processed as above, producing 9,101 passing cell profiles with a relatively low TSSe of 5.6, suggesting tissue preservation may be a factor. We next leveraged the s3-ATAC profiles, the ATAC modality from the sciMET+ATAC assay, and an additional annotated reference dataset of ~37,000 cells, enriched for NeuN(−) (~85%)[34] to produce integrated clustering and visualization, using the annotations from the reference atlas to assign cell types to each cluster (Figures 5D, 5E, and S3A).

We next processed the DNA methylation side, producing cell groupings similar to the assigned cell types from the

ATAC modality (Figure 6A). The methylation modality was combined with our previous sciMETv3 dataset produced on the same individual, which produced substantial overlap except for a single cluster that was able to be filtered out using our doublet detection model, suggesting elevated noise in the form of ruptured nuclei and ambient chromatin crosstalk in the dataset compared to the unimodal sciMETv3 workflow (Figures S3B–S3D). This form of noise is likely to occur more frequently when nuclei undergo two rounds of tagmentation versus a single tagmentation in either unimodal assay. We then leveraged the cluster identities from the unimodal dataset, as annotated in Figure 4, and performed label transfer to the sciMET+ATAC cells (Figure 6A). Using the ATAC and MET cell type classes, we next compared cross-modality assignments, which were largely concordant, including when leveraging the higher-granularity methylation-based clusters, with the exception of modest crosstalk between oligodendrocyte and oligodendrocyte precursor (OPC) cell populations (Figure 6B).

Paired ATAC and genome-wide DNA methylation enables the assessment of both open and closed chromatin for DNA methylation status, as opposed to methods that conduct bisulfite conversion only on ATAC-derived reads, providing insight into the regulatory status of loci across all cell types and not just those that exhibit open chromatin. To assess these interactions, we leveraged the methylation-based cell typing to produce aggregated ATAC tracks, producing distinct cell-type-specific accessibility patterns at marker genes (Figure 6C). We then assessed ATAC peaks called from the data for methylation status across cell types, splitting out the ATAC peaks by promoters and enhancers (Figure 6D). Between these categories, methylation was less variable at promoter regions, with nearly all cell types exhibiting hypomethylation. This low-variance hypomethylation population was present in the enhancer peak set, albeit only for a minority of peaks, with the large majority exhibiting higher methylation variance where a majority of cell types exhibited hypermethylation. This observation is consistent with previous studies that have shown that enhancers with intermediate DNA methylation have the highest tissue and developmental variance.[35–39]

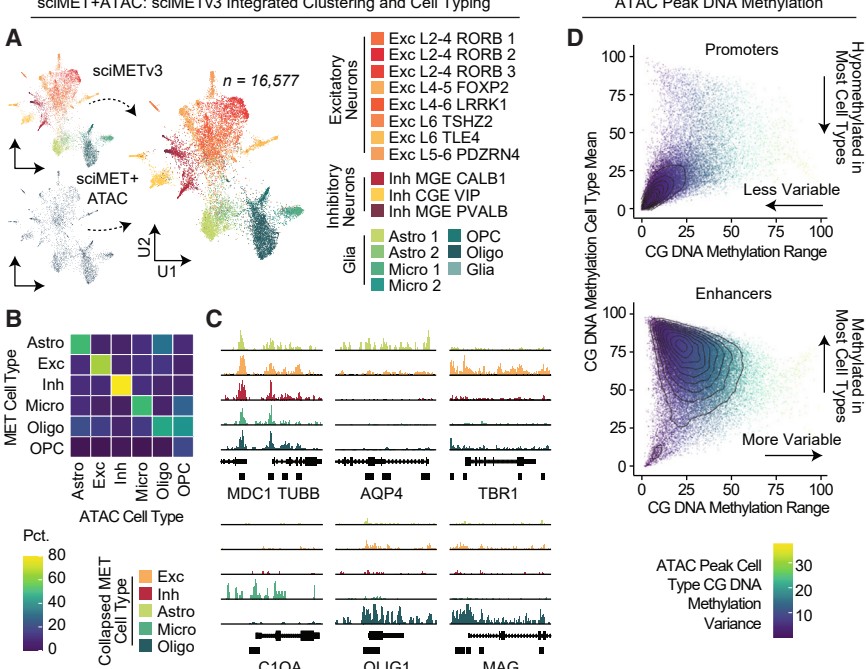

**sciMET+ATAC: sciMETv3 Integrated Clustering and Cell Typing**

**A** sciMETv3  n = 16,577

sciMET+ ATAC

U2  U1

Excitatory Neurons:
- Exc L2-4 RORB 1
- Exc L2-4 RORB 2
- Exc L2-4 RORB 3
- Exc L4-5 FOXP2
- Exc L4-6 LRRK1
- Exc L6 TSHZ2
- Exc L6 TLE4
- Exc L5-6 PDZRN4

Inhibitory Neurons:
- Inh MGE CALB1
- Inh CGE VIP
- Inh MGE PVALB

Glia:
- Astro 1
- Astro 2
- Micro 1
- Micro 2
- OPC
- Oligo
- Glia

**B**
MET Cell Type: Astro, Exc, Inh, Micro, Oligo, OPC
ATAC Cell Type: Astro, Exc, Inh, Micro, Oligo, OPC

Pct.: 80 60 40 20 0

Collapsed MET Cell Type:
- Exc
- Inh
- Astro
- Micro
- Oligo

**C** MDC1 TUBB   AQP4   TBR1
C1QA   OLIG1   MAG

**ATAC Peak DNA Methylation**

**D** Promoters

Hypomethylated in Most Cell Types

Less Variable

CG DNA Methylation Cell Type Mean vs CG DNA Methylation Range

Enhancers

Methylated in Most Cell Types

More Variable

CG DNA Methylation Range

ATAC Peak Cell Type CG DNA Methylation Variance: 30 20 10

**Figure 6. sciMET+ATAC methylation clustering and joint property analysis**

(A) DNA methylation modality integrated with sciMETv3 reference cells from the same individual and colored by cell type.

(B) Cross-modality cell type concordance.

(C) ATAC profiles of marker genes split by DNA-methylation-derived cell type.

(D) Called ATAC peaks at promoter regions exhibit less CG methylation variability between cell types versus putative enhancer peaks with higher cell type specificity.

## DISCUSSION

In recent years, large-scale DNA methylation atlases have enabled valuable insight into the contributions of DNA methylation to the regulatory landscape.[6–9] These studies were achieved using extensive automation and other resources well beyond the means of a typical research program. Here, we describe sciMETv3, a robust technology for the production of atlas-scale single-cell DNA methylation datasets capable of delivering library sizes in the hundreds of thousands of cells. The primary advantage of sciMETv3 over technologies leveraged to produce previous atlas-scale datasets is that a ready-to-sequence library containing comparable cell throughput (e.g., 500,000 cells) can be produced by a single individual over a few days without any special equipment and utilizing a total of seven 96-well plates plus a single 96-well column cleanup. In contrast, prior atlas technologies and similar methods that require a single well per cell require a minimum of 1,302 or 5,208 microwell plates (384-well or 96-well, respectively) to be processed at each step until pooling can be carried out, likely taking several months and substantial other resources to achieve the same scale. Notably, the prior atlas-scale studies, as part of the NIH BRAIN Initiative, sequenced each library to a greater depth per cell than we demonstrate here; however, we achieved our desired cell type clustering granularity at the depth produced, though additional sequencing could be carried out to increase depth to a level approaching that of the BRAIN Initiative studies. Further gains in coverage could also be achieved by leveraging our previously described sciMETv2 variant that leverages linear amplification (sciMETv2.LA) to append the reverse adapter, which can provide as much as a 10-fold library complexity increase, albeit at the cost of a longer protocol.[11] Such approaches are only relevant when very high per-cell coverage is desired, at which point the primary factor lies in sequencing cost.

Beyond the cell throughput advances, we demonstrate that sciMETv3 is compatible with capture-based techniques which allow for a reduced amount of sequencing to produce robust cell type clustering. The reduction in required sequencing is commensurate with the fold-enrichment of the targeted capture regions, which was demonstrated for both human brain and peripheral blood mononuclear cell (PBMC) tissues when focusing on CG methylation. In brain, where CH methylation is abundant and genome wide, as opposed to the enrichment at regulatory loci observed with the CG context, the gains from capture are less pronounced, reducing its utility. Our assessment of sciMETv3 with capture allowed for approximately 6,000 single-cell libraries to be multiplexed within a single capture reaction without a substantial reduction in the on-target capture rate, achieving a 6.2-fold enrichment. Notably, the capture workflows produce sufficient off-target coverage to provide genome-wide methylation calls when cells are aggregated at the cluster level, mitigating the limitation of capture techniques where non-targeted regions are missed.

The higher cell counts in the final indexing stage of sciMETv3 (~600–1,000) over its predecessor, sciMETv2 (15–60), makes alternative means of C-to-T conversion viable, including EM-seq methods. We demonstrate the use of enzymatic conversion on sciMETv3 libraries that produced a slightly larger fragment length, which is likely due to the gentler treatment of the DNA by enzymatic steps versus the harsh chemical treatment with sodium bisulfite. Resulting libraries produced comparable methylation profiles and did not exhibit any bias in clustering or cell type proportions when compared to standard bisulfite-based conversion libraries. This result was confirmed by leveraging enzymatic conversion for libraries prepared using protocols for Ultima Genomics sequencing, where results were again indistinguishable from bisulfite-based converted libraries. However, we observed over-conversion of sequencing adapters that impeded Illumina sequencing, which was not a factor using the Ultima platform due to the use of alternate primer regions.

We then demonstrate the production of a large-scale dataset produced from four human cortex specimens (middle frontal gyrus). Libraries were sequenced on either an Illumina NovaSeq 6000 or Ultima Genomics UG100 instrument. Notably, the single-end long-read-length nature of the UG100 instrument allows for minimal over-sequencing of internal bases within library fragments that get sequenced twice using paired-end sequencing where paired reads overlap. Achieving a longer fragment length could mitigate this observation, though even with enzymatic conversion methods, a substantial number of fragments would exhibit overlapping coverage using the paired 200 bp sequencing format that we used in this study. When evaluating the performance of each sequencing platform with respect to read recovery and alignment rates, methylation call percentages, and cell type compositions across four samples, each sequencing platform is functionally equivalent from a data quality standpoint. The integration of all cells sequenced from this preparation from both platforms generated a high-resolution atlas of cell types in human middle frontal gyrus, producing genome-wide maps of methylation profiles for each identified cell type along with HMRs.

Finally, we leverage a double-tagmentation workflow using two rounds of indexed Tn5 complexes with methylated adapters and an intervening nucleosome-disruption step. This workflow, sciMET+ATAC, enables the first tagmentation index to be leveraged for assessing chromatin accessibility and the combination of both to be used as genome-wide DNA methylation. Unlike NOMe-based techniques that encode accessibility using artificial GpC methylation, our strategy enriches for accessible loci using ATAC-based workflows, providing enough information per cell to enable cell-level ATAC analysis as opposed to cluster level. Additionally, the technique is applicable to brain and embryonic stem cells, where GpC methylation is common due to the presence of noncanonical CH-context methylation genome wide,[39,40] complicating NOMe-based approaches. Overall, the data quality of sciMET+ATAC is lower for each modality than when each is performed on its own, as represented by a lower TSSe value in the ATAC modality and the presence of noise in the methylation modality likely due to nuclei rupture and elevated ambient chromatin fragments. However, the use of tailored QC filtering allowed for distinct cell type identification, bolstered by integration with reference sciMETv3 cells from the same individual. Similarly, the ATAC modality integrated with an s3-ATAC dataset which we generated for this study using a novel nanowell-chip-based implementation of the s3-ATAC workflow, as well as with an annotated reference dataset. Integration with these two ATAC datasets enabled direct cell type label transfer to the sciMET+ATAC cells. Notably, the DNA methylation modality was able to produce a higher granularity of neuronal clusters, likely due to the richness of CH methylation across the genome and the high information content produced using the sciMET assay. Taken together, we believe that the sciMET+ATAC workflow will be a valuable for profiling a portion of cells in addition to the sciMETv3 workflow to bridge between datasets and facilitate cross-modality integration and cell type assignment as opposed to serving as a standalone dataset due to the reduced TSSe and elevated ambient chromatin noise.

## Limitations

In its presented form, sciMETv3 produces large-scale datasets yet requires a relatively large number of starting cells or nuclei, on the order of 250,000 to 1 million, depending on the fragility of the nuclei, which impacts the survival during nucleosome disruption. This makes the method most suitable for applications where the starting material is not limited, as opposed to low-input samples. A related limitation is that the large-scale library produced will likely exceed what can reasonably be sequenced, yet the preparation still requires a high cell input even if only a subset of final wells (∼750–1,000 cell libraries each) are desired.

Another limitation of sciMETv3 is the fragment size, which is lower than methods that deposit cells into individual wells of a plate. This is due to two steps that can fragment DNA: the conversion process, shared by all methods, and the upfront tagmentation step that is unique to sciMET technologies. The reduced fragment size limits the ability to perform allele-specific methylation analysis. Deploying enzymatic conversion increases the fragment length, which may improve such analyses. However, the undesired conversion of methylated bases in the sequencing adapter makes this approach viable on specific sequencing platforms. Addressing this would require alternative protection modifications that are extremely expensive or a complete adapter redesign.

Finally, we presented sciMET+ATAC, which generated usable data for each modality yet has several key limitations. The first is the challenge of performing multiplexed nucleosome disruption on sets of nuclei that already experienced *in situ* tagmentation for the ATAC modality. After initial tagmentation, nuclei are more fragile, leading to higher loss and requiring special care when handling. This is compounded by the need to perform the nucleosome disruption in a multiplexed form versus the individual 1.5 mL tube per sample that can be processed for standard sciMET assays. In this work, we presented an 8-plex nucleosome disruption, which meant 8-plex for the initial round of cell barcoding. In theory, this could be expanded to the full 96-plex, yet the practicality of performing plate-scale nucleosome disruption is likely prohibitive.

## RESOURCE AVAILABILITY

### Lead contact

Further information and requests for resources and reagents should be directed to and will be fulfilled by the lead contact, Andrew C. Adey (adey@ohsu.edu).

### Materials availability

All materials used in this study are readily available from commercial vendors.

### Data and code availability

All raw and processed data for this study are available in public repositories with unrestricted access. Raw data can be accessed from the NCBI Sequence Read Archive (SRA) under project accessions SRA: PRJNA1126272 (sciMETv3) and SRA: PRJNA1134352 (sciMET+ATAC). Processed data can be accessed from the NCBI Gene Expression Omnibus (GEO) under accessions GEO: GSE273592 (sciMETv3) and GEO: GSE272699 (sciMET+ATAC). All analyses were performed using publicly available software. DNA methylation analysis was carried out using Amethyst.[18]

## ACKNOWLEDGMENTS

Funding for this work was provided by NIH BRAIN Initiative RF1MH128842 and a Silver Family Foundation Innovator Award to A.C.A. Sequencing on the Ultima Genomics UG100 instrument was carried out at Ultima Genomics on libraries provided by the Adey Lab. We would like to thank Doron Lipson, PhD, Ariel Jaimovich, PhD, Alix Cruise, PhD, and Mirna Jarosz, PhD, of Ultima Genomics for facilitating the collaboration and providing sequencing data.

## AUTHOR CONTRIBUTIONS

R.V.N., B.L.O., and A.C.A. conceived the sciMETv3 and sciMET+ATAC technologies. R.V.N. performed all sciMETv3 and sciMET+ATAC preparations with assistance from B.L.O. B.L.O. performed the s3-ATAC preparation and developed the nanowell-based s3-ATAC workflow. S.N.A. performed the capture experiments. L.E.R., B.L.O., and A.C.A. performed the data analysis. Z.S. and N.I. performed Ultima Genomics sequencing and primary data processing of those data. A.C.A. supervised all aspects of the study and wrote the manuscript with input from all authors.

## DECLARATION OF INTERESTS

A.C.A. is an author of one or more patents that pertain to sciMET technology and an advisor to Scale Biosciences. This potential conflict is managed by the office of research integrity at OHSU. Z.S. and N.I. are employees of Ultima Genomics.

## STAR★METHODS

Detailed methods are provided in the online version of this paper and include the following:

- KEY RESOURCES TABLE
- EXPERIMENTAL MODEL AND STUDY PARTICIPANT DETAILS
- METHOD DETAILS
  - Tissue homogenization and nuclei isolation
  - Nucleosome disruption
  - Barcode 1: Tagmentation
  - Barcode 2: ligation
  - Post-ligation & dilution
  - Bisulfite conversion (BSC), cleanup, and adapter ligation
  - Enzymatic conversion, cleanup, and adapter ligation
  - Barcode 3: Indexing PCR
  - Ultima indexing PCR
  - sciMETv3 capture
  - sciMETv3+ATAC
  - s3-ATAC iCell8 loading protocol
- QUANTIFICATION AND STATISTICAL ANALYSIS
  - Read processing
  - Alignment and methylation call extraction
  - DNA methylation analysis using amethyst
  - s3-ATAC sample extraction and barcoded tagmentation
  - s3-ATAC analysis

## SUPPLEMENTAL INFORMATION

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

## STAR★METHODS

### KEY RESOURCES TABLE

| REAGENT or RESOURCE | SOURCE | IDENTIFIER |
|---|---|---|
| **Biological samples** | | |
| Healthy adult BA46 brain tissue | National Institute of Health | UMBN#6596 UMBN#6926 UMBN#6996 UMBN#6998 |
| Healthy adult frontal cortex brain tissue | Oregon Health and Science University | 2972 A2 |
| Healthy p8 mouse right hippocampus brain tissue | Oregon Health and Science University | N/A |
| **Chemicals, peptides, and recombinant proteins** | | |
| HEPES, pH 7.5 | Sigma-Aldrich | H4034 |
| MgCl | Sigma-Aldrich | M8226 |
| NaCl | Fisher Scientific | M-11624 |
| IGEPAL | Sigma-Aldrich | I8896 |
| Tween 20 | Sigma-Aldrich | P7949 |
| Formaldehyde | Fisher Scientific | PI28906 |
| Glycine | Sigma-Aldrich | G8898-500G |
| UltraPure Sodium Dodecyl Sulfate | Invitrogen | 15525–017 |
| DMF 16% | Sigma-Aldrich | D4551 |
| Taps pH 8.5 | Sigma-Aldrich | T5130 |
| Proteinase K | Qiagen | 19086 |
| Buffer EB | Qiagen | 19131 |
| VeraSeq ULtra DNA Polymerase | Enzymatics | P7520S |
| T4 DNA Ligase | New England Biolabs | M0202M |
| T4 Polynucleotide Kinase | New England Biolabs | M0201S |
| ET-SSB | New England Biolabs | M2401S |
| T4 RNA Ligase Buffer | New England Biolabs | B0216L |
| PEG 8000 (50% w/v) | New England Biolabs | B0216L |
| 1,3-Propanediol | Merck | 504-63-2 |
| ATP 100 mM | ThermoScientific | R0441 |
| Protease Inhibitor | Roche | 11873580001 |
| KOAc | Sigma-Aldrich | P1190 |
| L-Desulphonation Buffer | Zymo Research | N/A |
| Zymo-Spin I-96 Plate (Shallow well) | Zymo Research | C2004-SW |
| **Critical commercial assays** | | |
| EZ-DNA Methylation Direct Kit | Zymo Research | N/A |
| Qubit 1x dsDNA HS Assay Kit | Invitrogen | Q33231 |
| High Sensitivity D1000 ScreenTape | Agilent | 5067–5584 |
| High Sensitivity D1000 Sample Buffer | Agilent | 5067–5603 |
| Scale Bio Single Cell Methylation Kit v1.1 | Scale Biosciences | 1064641 |
| **Deposited data** | | |
| sciMETv3 Raw Data | SRA | SRA: PRJNA1126272 |
| sciMETv3 Processed Data | GEO | GEO: GSE273592 |
| sciMET+ATAC Raw Data | SRA | SRA: PRJNA1134352 |
| sciMET+ATAC Processed Data | GEO | GEO: GSE272699 |

## Technology

CellPress

*Continued*

| REAGENT or RESOURCE | SOURCE | IDENTIFIER |
|---|---|---|
| Oligonucleotides | | |
| /5AmMC6/*G*AGCACACTCTGAACTCCAGTC*A*/3ddC/ | Integrated DNA Technologies | mod-i7-splint |
| /5Phos/AGATCGGAAGAGCGTCGTGTAGGGA AAGAGTGT/3AmMO/ | Integrated DNA Technologies | SL_TruSeq_i5_rc |
| /5AmMC12/ACACTCTTTCCCTACACGACGCTCTTCCG ATCT(H:33330034)(H)(H)(H)(H)(H)(H)(H)(H)/3AmMO/ | Integrated DNA Technologies | SL_TSi5splint_H10 |
| Software and algorithms | | |
| unidex | Adey Lab | https://github.com/adeylab/unidex |
| amethyst | Adey Lab | https://github.com/lrylaarsdam/amethyst |
| premethyst | Adey Lab | https://github.com/adeylab/premethyst |

## EXPERIMENTAL MODEL AND STUDY PARTICIPANT DETAILS

All tissue specimens were obtained from the NIH NeuroBiobank as a part of the NIH BRAIN Initiative Cell Atlas Network (BICAN) collections which are consented for open data release. Ethical oversight was carried out by the OHSU Institutional Review Board.

## METHOD DETAILS

### Tissue homogenization and nuclei isolation
The brain tissue was dounce homogenized using cold NIB-Hepes buffer (10 mM HEPES, pH 7.5, 3 mM MgCl$_2$, 10 mM NaCl, 0.1% IGEPAL (v/v), 0.1% Tween 20 (v/v), 1x protease inhibitor) as in Nichols et al. 2022. The cell suspension was then spun down (5 min, 500xg, 4°C). The pellet was then resuspended in NIB-Hepes for nuclei quantification.

### Nucleosome disruption
Nuclei were quantified using a K2 Cellometer. Samples were separated into 1 million nuclei aliquots. Each aliquot was taken through the ScaleBio DNA Methylation Kit protocol for fixation and nucleosome disruption following manufacturer's instructions. Afterward nuclei were spun down at room temperature and resuspended in NIB-H. Aliquots were then recombined and quantified.

### Barcode 1: Tagmentation
We tagmented 10,000–50,000 nuclei per well in a 96-well plate using Tn5 loaded with adapters containing all methylated cytosines (ScaleBio Part No: 941770). Each well contained 10 μL tagmentation buffer (ScaleBio Part No: 941788). The plate was incubated at 55°C for 15 min and then placed on ice. All wells were pooled and put into a 5 mL tube. 2 mL cold NIB-H was added, and the mixture was spun down at 500xG 4C for 5 min. The supernatant was removed. The mixture was washed with cold NIB-H + 3 μL BSA, spun down, and the supernatant was removed. The nuclei were then resuspended in 110 μL cold NIB-H, quantified, and used for *in situ* ligation.

### Barcode 2: ligation
To the 110 μL of nuclei, the following was added: 33 μL 10X Polynucleotide Kinase Buffer, 33 μL 10 mM ATP, 22 μL dH$_2$O and 132 μL T4 Polynucleotide Kinase. The mixture was mixed by pipetting and distributed to a plate at 3 μL per well. The plate was incubated at 37°C for 30 min and then placed on ice. 2 μL of 15 μM ligation barcode oligos were added to each well of the plate (IDT, Table S1). The following was then added to each well of the plate: 6.2 μL 2X StickTogether Buffer, 0.3 μL 100 μM v3 ligation splint and 1.5 μL T7 DNA Ligase. In other versions/experiments, the nuclei were kept in a 1.5 mL tube for the PNK 37°C incubation, after which the ligation master mix was added and the nuclei distributed to the plate containing the 96 ligation barcodes. The plate was incubated at 25°C for 1 h and then placed on ice and allowed to cool fully. A full list of ligation oligo sequences can be found in Table S1.

### Post-ligation & dilution
All wells were pooled into a 5 mL tube. 3 mL NIB-H and 3 μL BSA were added. Nuclei were then spun down at 4°C 500xG for 5 min. The supernatant was removed. 3 mL NIB-H (with no protease inhibitors) was added. The tube was then spun down at 4°C 500xG for 5 min and resuspended in 100 μL NIB-H (no protease inhibitors). Nuclei were quantified and diluted to 750 nuclei per μL and 1 μL was added to each well of the final plates for bisulfite conversion using the ScaleBio Methylation Kit Met Bisulfite Conversion Module (Part No: 943631). Final plates or wells that used enzymatic conversion had 1 μL Qiagen Protease and 1 μL 90 mM Tris. The plates were spun down briefly and frozen at −20°C.

### Bisulfite conversion (BSC), cleanup, and adapter ligation

The plates for bisulfite conversion were defrosted and spun down briefly to collect the liquid to the bottom of the wells. Plates were then incubated at 50°C for 20 min to digest the nuclei and reverse cross-links. Bisulfite conversion, cleanup and reverse adapter ligation was carried out using manufacturers protocols for the ScaleBio Single-Cell DNA Methylation kit (ScaleBio Part No: 943631, 944302 and 944376).

### Enzymatic conversion, cleanup, and adapter ligation

Enzymatic conversion was carried out using the NEBNext Enzymatic Methyl-seq Conversion Module. Plates were spun down and then incubated at 55°C for 15 min and 72°C for 20 min to inactivate the Qiagen Protease. Afterward, the manufacturer's protocol was followed for enzymatic conversion. Final elution was done using 10 μL EB and then carried through the ScaleBio Single-cell DNA Methylation Kit (ScaleBio Part No: 944376) workflow for adapter ligation.

### Barcode 3: Indexing PCR

The indexing PCR was performed with the following recipe for each well of a 96-well plate: 10 μL 5X VeraSeq GC Buffer, 2 μL 10 mM dNTPs, 1.5 μL VeraSeq ULtra Polymerase, 24 μL dH$_2$O, 0.5 μL EvaGreen 100X and 1 μL 1 μM i7 Flow Cell primer for a total volume of 39 μL. 1 μL of barcoded i5 primers was added separately to each well. A full list of primers can be found in Table S1. The plate was mixed and placed on a qPCR with the following thermal conditions: 98°C initial denaturation for 30 s, 98°C for 30 s, 57°C annealing for 20 s, 72°C extension for 20 s, 72°C plate read for 10 s (these last 4 steps were cycled until exponential amplification was seen). After PCR, 10 μL of each well was pooled and the pool was column cleaned and SPRI cleaned with equal volume of product to SPRI beads. The resulting library was quantified using Qubit and TapeStation. Libraries were sequenced on an Illumina NextSeq 2000 or Illumina NovaSeq 6000.

### Ultima indexing PCR

For Ultima-compatible libraries, indexing PCR was carried out as above but substituting primers that ensure all indexes are on the same side of the molecule and that contain the Ultima Genomics outermost amplification and sequencing primers. A full list of primers can be found in Table S1. The final plate was pooled and sequenced on an Ultima Genomics UG100 instrument using six wafers.

### sciMETv3 capture

We pooled an 8-strip of sciMETv3 library in a volume of 16 μL of water. We performed capture with standard blockers and 300ng of library material. In a tube we combined 4 μL methylome panel (Twist Human Methylome Panel, Twist Bioscience, 105520), 8 μL Universal Blockers (also known as standard blockers, Twist Biosciences, 100578), 5 μL Blocker Solution (Twist Biosciences, 100578), 2 μL Methylation Enhancer (Twist Biosciences, 103557) and 1 μg of library in a volume of 7 μL in a 1.5 mL Eppendorf tube. Tubes were dried down on low heat in a speed-vac for 15 min and checked every 15 min for about an hour.

A thermal cycler was programmed as follows: 95°C hold/95°C 5'/60°C hold (lid 85°C). While the empty thermocycler was held at the first 95°C step, 20 μL of 65°C Fast Hybridization Mix (Twist Biosciences, 104180) was added to tubes with dried down panel, library and blockers. The mixture was solubilized for an additional 5 min before transferring to a 0.2 mL PCR tube. 30 μL of Hybridization Enhancer (Twist Biosciences, 104180) was added, the tube was pulse-spun and then transferred to the hot thermal cycler which was then allowed to continue past the initial 95°C hold. The reaction was hybridized for 16 h at the 60°C hold step to account for the large size of the methylome panel. Subsequent washing and PCR amplification was carried out according to manufacturer's protocol, using a 63°C wash temperature.

### sciMETv3+ATAC

Nuclei were isolated in the same way as above. We tagmented 100,000–500,000 nuclei per well in an 8-strip using Tn5 loaded with adapters containing all methylated cytosines (ScaleBio Part No: 941770). Each well contained 10 μL tagmentation buffer (ScaleBio Part No: 941788). The 8-strip was incubated at 55°C for 10 min with 400 RPM shaking and then placed on ice. Each well was transferred to its own 1.5 mL tube where they were fixed and nucleosome disrupted using the same protocol as for the full sciMETv3 version. After nucleosome disruption it is important to remove all of the supernatant without disturbing the pellet. For the second tagmentation a new set of 8 barcodes was used and the nuclei were tagmented using the same recipe as above. They were also tagmented at 55°C for 10 min with 400 RPM shaking and then placed on ice. All wells were then pooled and carried through all post-tagmentation steps of the sciMETv3 protocol. Final plates had 90 nuclei diluted per well.

### s3-ATAC iCell8 loading protocol

For this sample, all additions were in a 36 × 36 well format. Volumes should be doubled if running the protocol in 72 × 72 well mode to account for the additional wells. LNA/SDS mix[32] was distributed into a 350v iCell8 chip (TakaraBio) at 35 nL per well. The chip was blotted, capped with RC film (TakaraBio) and centrifuged 10:00 at 2500 rcf 4°C between every step unless otherwise specified. 50 nL diluted cells were then added, followed by incubation at 65°C for 10:00° and 72° for 10:00 (note: exact temperature settings for the

modified BioRad T-1000 thermocycler included with the iCell8 were based on the conversion tables in Appendix G: Designing Thermocycler Programs, contained within the iCell8-CX User Manual).

Following nuclear lysis, we dispensed the Quench/Linear Extension mix[32] at 50 nL per well. The chip was then placed in the BioRad TC-1000 for the Adapter Extension step. Adapter switching conditions were: initial extension of 72°C for 10 min, initial denaturation at 98°C for 30 s, then 10 cycles of 98°C for 10 s, 59°C for 20 s, and 72°C for 1 min, followed by a 72°C final extension for 1 min and cooling to 10°C hold.

35 nL each i7 TrueSeq and i5 Nextera barcoded PCR primers (15 μM) were added to each well, then 100 nL PCR Master-mix[32] was added to each well (note: this can be added in 1x100 nL dispense for 36x36 mode, but must be added in 2, 50 nL dispense steps for 72x72 well mode). Final amplification conditions were: 98°C for 45 s, then 13 cycles of 98°C for 15 s, 57°C for 30 s, and 72°C for 30 s, finishing with a 72°C final extension for 5 min.

After extension, the PCR product was extracted from the iCell8 chip by centrifugation with the provided funnel from TakaraBio, and a 300 μL aliquot was SPRI cleaned with a 1:1 sequential SPRI clean (sequentially adding 100 μL 3 times to the aliquot with 2 min binding time between additions to improve the size selection effect), before being eluted in 30ul and quantified with Qubit DNA fluorometer HS kit (Invitrogen) and Aglient Tapestation D1000.

The purified library was sequenced on a NextSeq 2000 P3 kit, with the following cycle numbers: Read 1: 89 bp, Index 1: 10 bp, Index 2: 10 bp, Read 2: 129 bp.

## QUANTIFICATION AND STATISTICAL ANALYSIS

### Read processing

Raw sequence reads produced using Illumina instrumentation were carried through barcode demultiplexing using unidex (github.com/adeylab/unidex) to produce barcode-corrected read name paired fastq files. Reads were then taken through adapter trimming using 'premethyst trim' (github.com/adeylab/premethyst), which leverages Trim Galore (https://doi.org/10.5281/ZENODO.7598955). Sequence reads produced using the Ultima Genomics instrument were processed using the Ultima Genomics demultiplexing software to produce unaligned cram files containing the read with adapter bases trimmed and error-corrected indexes as a special field. These crams were then converted to fastq files with barcodes included within the read name for downstream compatibility.

### Alignment and methylation call extraction

Fastq files were aligned using the 'premethyst align' wrapper using default parameters which leverages BSBolt.[41] Aligned bam files were deduplicated using 'premethyst rmdup' and then methylation call files were generated using 'premethyst extract', including a lenient minimum read count threshold of 10,000 since cells are later filtered using more stringent parameters at subsequent analysis steps. Call files were then packaged into h5 calls files using 'premethyst export'.

### DNA methylation analysis using amethyst

Cell metadata 'cellInfo' files produced from 'premethyst extract' along with methylation call h5 files were used to generate an Amethyst analysis object using amethyst (github.com/lrylaarsdam/amethyst)[18] and then filtered to include cells meeting minimum cytosine coverage levels (1M for atlas dataset, 500k for other datasets). An hg38 reference annotation file was added for gene-level coordinates with the 'makeRef()' function. Site-level information in the h5 files were cataloged by chromosome using 'indexChr'. Window methylation matrices were then generated with 'makeWindows', both for CG using metric = 'score' and CH using metric = 'percent'. For the large-scale datasets produced using the Illumina NovaSeq 6000 and Ultima Genomics UG100 instruments, 100 kbp windows were leveraged, expanding to 200 kbp windows for all other smaller-scale datasets. We then estimated the number of IRLBA dimensions to calculate for the CG and CH contexts using 'dimEstimate()' followed by producing an IRLBA matrix using the specified number of recommended dimensions for each respective context using 'runIrlba()'. Effects of coverage bias on the irlba matrix were mitigated with 'regressCovBias()'. From the result, distinct groups were identified with the Rphenograph-based 'runCluster()' function and umap coordinates were projected using 'runUmap()'. Cell type identification of the resulting clusters was performed based on the consensus of the following modalities: mCG patterns over canonical marker genes using amethyst visualization functions 'histograM()' and 'heatMap'; mCH levels over canonical marker genes using functions 'dotM' and 'dimM'; and correlation of mCH levels over subtype-specific gene subset[6] to a reference atlas produced by the Ecker Lab.[7] Integration of the Illumina and Ultima datasets was carried out using Harmony performed on the irlba matrix. Cell type annotation for the sciMET+ATAC dataset for the methylation modality was performed by integration of the sciMETv3 Illumina dataset for the same individual, leveraging the previously-annotated cell types to label-transfer to the sciMET+ATAC cells using the amethyst function 'clusterLabelTransfer()'.

### s3-ATAC sample extraction and barcoded tagmentation

Frozen human brain tissue ID: 6996 was minced on dry ice and added to a Dounce homogenizer on ice along with cold 2 mL NIBH, containing fresh protease inhibitors. The tissue was homogenized with 7 strokes with the 'A' pestle, incubated for 10:00 on ice, then treated with 7 strokes with the 'B' pestle. The lysate was then filtered through 70 μm and 40 μm cell strainers (pluriSelect 43–50070 and 43–50030) and centrifuged at 500rcfr for 6:00 to remove extranuclear debris. The pellet was resuspended in 0.5 mL NIBH and counted on a Revvity K2 cellometer.

We performed tagmentation by adding 300 μL 4x TD Buffer and 12 μL 1M D-glucosamine (Sigma Aldrich), and an additional 388 μL NIBH to the nuclei for a final volume of 1.2 mL. We then distributed the nuclei into a 96-well PCR plate, at 10 μL per well, before adding 1.5 μL of barcoded tn5 (ScaleBio)[32] and tagmenting at 55°C for 15 min. The plate was then transferred to ice and incubated 5:00 before pooling the nuclei into a 5 mL tube and adding 3mL NIBH. The nuclei were then centrifuged for 6:30 at 500 rcf and washed with 3 mL NIBH plus 3uL 100 mg/ml BSA. After washing, the nuclei were resuspended in 100 μL, and counted on the K2 cellometer. Nuclei were diluted to 340 nuclei/μL for loading on the iCell8.

### s3-ATAC analysis

Sequencing data were demultiplexed with unidex and aligned with bwa mem.[42] The file was sorted by cell barcode, PCR duplicates were removed, and a custom python script was used to check the BAM file header for errors and add the cell barcode to the 'CB' BAM tag for each read to allow for faster ingest with SnapATAC2. A fragments file was created using SnapATAC2's make_fragment_-file function, and then an AnnData object was created with the import_data function with default parameters except for setting sorted_by_barcode to True. QC plots (fragment distribution and TSS enrichment) were generated as recommended in the SnapATAC2 documentation, and the dataset was filtered to remove cells with TSS-enrichment less than 5. Feature selection, dimensionality reduction, and clustering were all performed according to the SnapATAC documentation's recommended settings.

For cell type assignment, we used the HGAP dataset[34] and co-processed it with the human brain data described above, as well as with the ATAC data from the MET+ATAC coassay, using the same process described above, with the addition of removing batch effects with Harmony.[17] After clustering, the cell type information from the HGAP data was used to assign an implied cell types to the new datasets. As an additional validation, we use scanpy's tracksplots function to plot the accessibility of various brain cell type marker genes across the different clusters. The results were concordant with the annotation lift-over.

