## [Document S2. Transparent peer review records for Nichols et al · Cell Genomics]

Atlas-scale Single-cell DNA Methylation Profiling with sciMETv3

Author list

Ruth V. Nichols , Lauren E. Rylaarsdam, Brendan L. O'Connell, Zohar Shipony, Nika Iremadze, Sonia N. Acharya, Andrew C. Adey

Summary

Initial submission:	Received : September 3 rd 2024
	Scientific editor: Judith Nicholson
First round of review:	Number of reviewers: 2
	Revision invited : October 11 th 2024
	Revision received : October 25 th 2024
Second round of review:	Number of reviewers: 2
	Accepted : November 26 th 2024
Data freely available:	Yes
Code freely available:	Yes

This transparent peer review record is not systematically proofread, type-set, or edited. Special characters, formatting, and equations may fail to render properly. Standard procedural text within the editor's letters has been deleted for the sake of brevity, but all official correspondence specific to the manuscript has been preserved.

Referees' reports, first round of review**Reviewer 1**

Could you please comment on the following?-

1. Sequencing accuracy- Illumina sequencing was likely paired end, while that on Ultima was single-end sequencing. Did you see comparable accuracy in sequencing or base calling?
2. You likely needed to make the libraries compatible with Ultima. Usually the primers that are used to convert the libraries (at least in case of 3'-scRNA-seq libraries) bind the R2 region prior to the index sequence. Was that the case here as well? If yes, was a new index added to individual libraries and demultiplexing was done using those newly added indices?
3. Do you notice any bias towards any particular allele using sciMETv3? Or, can you potentially analyze the data at the allelic level?

Reviewer 2

DNA methylation is an important epigenetic modification known to control cell-type specific gene expression but is challenging to study in single cells. This manuscript addresses this challenge by presenting several modifications to a pre-existing technique. The updated method, called sciMETv3, incorporates additional index sequences into the library preparation strategy to increase the number of cells that can be studied in a single experiment (theoretically up to 10 million). The authors also show that sciMETv3 is compatible with capture-based enrichment methods, two sequencing platforms and parallel analysis of chromatin accessibility. Together, these modifications make sciMETv3 suitable for implementation in cell atlas projects.

The combinatorial indexing approach used in sciMET has several advantages over similar techniques already used in atlas-scale studies. For example, library preparation is performed on pools of nuclei as opposed to individual cells, eliminating the need for sophisticated robotic platforms. This manuscript would be strengthened by a more comprehensive comparison between sciMETv3 and alternative methods, highlighting the relative strengths and weaknesses of the different approaches.

The manuscript and figures are prepared to a very high standard and the text is succinct and easy to read. However, the extent to which sciMETv3 extends on past work from this group and others has been exaggerated by omitting key references, and by limiting the discussion to the strengths of sciMETv3.

Specific Comments

1. The authors present sciMETv3 as compatible with atlas-scale studies of DNA methylation, but do not pay due credence to alternative methods used in published atlas-scale studies.
 - a. Line 27-28: The authors claim that "robust and cost-effective techniques to produce atlas-scale datasets have not been realized for DNA methylation", but several atlas-scale datasets have already been produced (e.g. Liu et al Nature 2021 & Liu et al Nature 2023 624:336). Please rephrase this section of the text to acknowledge that atlas-scale DNA methylation datasets are already available.
 - b. In the Discussion, the authors should compare the strengths and weaknesses of their method to those of alternative methods.

2. Capture-based enrichment is used to reduce the sequencing demand of single-cell DNA methylation analysis, but limited detail is provided by the authors.
 - a. The authors list "Compatibility of sciMETv3 with capture techniques to reduce sequencing burden" as a highlight of this paper. This gives the impression that this is the first study to combine single-cell DNA methylation analysis with capture-based enrichment methods. However, the authors have recently published sciMET-cap in a July 2024 Genome Biology paper. The authors should revise this manuscript to appropriately describe past work and to highlight the novel contributions of this paper.
 - b. Line 278: The authors explain that capture-based techniques can reduce the sequencing demand of DNA methylation atlas studies, but it is not clear if this is unique to sciMETv3. Are other (plate-based) methods for single-cell analysis of DNA methylation also compatible with capture-based techniques?
 - c. Line 278: By how much is the sequencing demand reduced if one uses capture-based techniques?
 - d. Line 281: What is the off-target coverage associated with the capture-based techniques?

3. The authors present data to demonstrate that sciMETv3 is compatible with parallel analysis of chromatin accessibility by ATAC-seq; however, the resulting data is of reduced quality for both modalities. The authors acknowledge that the sciMETv3 data had "elevated noise" (line 257) and that the "transcription start site enrichment (TSSe) was relatively low" (line 235) in the ATAC-seq data. A possible alternative to this approach would be a Nucleosome Occupancy and Methylation (NOMe) sequencing approach in which a bacterial GpC methyltransferase is used to mark accessible chromatin prior to library preparation. Have the authors considered combining sciMETv3 with a NOMe-based approach to assess DNA methylation and chromatin accessibility in the same single cells?

4. Line 206-209: The authors explain that 12.1% and 28.6% of HMRs overlap promoters and enhancers, respectively. Is this overlap more or less than expected based on the number and genomic locations of promoters and enhancers? In other words, are HMRs enriched in enhancers?

5. Lines 271-274: The authors describe DNA methylation variance at promoters and enhancers defined using ATAC-seq data from the matched cell population. The results show that enhancers with intermediate levels of DNA methylation have the highest variance. This finding is consistent with several previous reports, which should be cited here.

6. Lines 257, 309 and Figure S3D: In the combined sciMETv3/ATAC analysis, there is a cluster of cells enriched for high doublet scores. The authors refer to this observation using the somewhat ambiguous term "noise". Can the authors explain why there may be a higher doublet rate in the multi-modal analysis? Line 256-258 should be revised to describe the observation in less ambiguous terms.

7. There is no information regarding ethics approvals in the Methods section. Was this study conducted with appropriate ethical oversight?

Minor Corrections, Suggestions and Queries

1. Line 61: Consider replacing unfamiliar word "amortizes".
 2. Figure 1E: It is confusing to have a legend symbol for "Mix" when there are no cells in this category. Consider removing this symbol from the graph.
 3. Line 428, 447: Use subscript for "2" in "MgCl₂" and "dH₂O"
 4. Line 429: "Nichols et al. 2022" is not in the reference list.
 5. Line 430, 442 and throughout: Degrees Celsius should be written as "°C".
 6. Line 444: "in situ" should be italicized
 7. Line 449: Please specify if the ligation barcodes are included in a kit.
 8. Line 497: Please check: "95°C hold / 95°C 5' / 60°C hold (lid 85°C). 20 µl of 65°C." What does 20 µl refer to? Is there an extra full stop?
 9. Line 519: Please add a citation for Trim Galore.
 10. Line 544: Please remove mid-word space in "using amethyst visualization functions".
-

Authors' response to the first round of review

Response to Reviewers:

We thank the reviewers for their detailed critique of our manuscript. Each reviewer provided important feedback that we believe better frames the work and provides improved motivation for the development of the sciMETv3 technology. Below we provide a point-by-point response, with original comments in black, and responses in blue. We also include snippets of the manuscript showing tracked changes, as well as altered portions of figures where appropriate.

Reviewers' Comments:

Reviewer #1:

Could you please comment on the following?

1. Sequencing accuracy- Illumina sequencing was likely paired end, while that on Ultima was single-end sequencing. Did you see comparable accuracy in sequencing or base calling?

For conversion-based sequencing (bisulfite or enzymatic), accuracy is somewhat challenging to assess due to the inability to discern conversion differences from sequencer accuracy. For purposes of sciMETv3, the key components for viability from a sequencing platform are index assignment, alignment rates and methylation calling. We discuss each below along with their respective challenges:

For both platforms index assignment rates were comparable regarding the percentage of reads being assigned to whitelisted cell barcode combinations (78.0% and 84.7% for Illumina and Ultima respectively). However, each platform leveraged a different index demultiplexing tool, either a custom tool for Illumina reads (unidex, referenced in the methods section) or the default index demultiplexer provided by Ultima Genomics, so there are likely differences due to software, though rates are still similar. Different tools were used because different index designs were leveraged that are optimized for each of the platforms. Also worth mentioning is that the total raw read yield for the Ultima sequencing was lower than expected (28.5 Bn raw reads for 6 wafers, vs an expected 32-48 Bn), meaning the instrument may have been slightly underloaded, which theoretically could provide an increase in read quality due to less crowding on the flowcell.

We next assessed mapping rates, where Illumina read pairs achieved an alignment rate of 94.3% compared to 93.7% for Ultima single end reads, suggesting comparable accuracy with respect to the ability to align. This is also difficult to draw much of a conclusion from, due to the potential selection bias imposed by the index demultiplexing. A more stringent index caller will likely select reads with a higher overall quality, as the index quality correlates closely with read quality. However, rates are still very close, making it reasonable to assume both are comparable. We now include these alignment rates in the Results section:

The first plate was sequenced on a single S4 flowcell of an Illumina NovaSeq 6000™ instrument using a paired 200 cycle kit, producing 7.18 billion raw read pairs after demultiplexing (78.0% yield) and an alignment rate of 94.3%. This resulted in 44,840 total cells called with a median of 1.82 million cytosines

And:

The plate sequenced using the Ultima Genomics UG100™ instrument was processed over six wafers, yielding a total of 28.5 billion raw reads, 24.13 billion after demultiplexing and trimming (included in the demultiplexing process, see methods) with an alignment rate of 93.7%. The increased read counts over the

Lastly, the methylation calling is comparable for the two platforms, though this can also be skewed by the conversion process itself. As shown in figure 3H (below) the global methylation rates in both CG and CH contexts are lower for Ultima for both the bisulfite and enzymatic preparations. It is not possible to discern which of the methods is correct with no ground truth with which to compare. In either case, both methods integrate cleanly, making the data produced essentially indistinguishable for the vast majority of applications.

Given the challenges and potential biases associated with each of these criteria, we elected to make the claim in the Discussion that with respect to sciMETv3 study goals, both platforms are functionally equivalent.

200 bp sequencing format that we used in this study. When evaluation the performance of each sequencing platform with respect to read recovery and alignment rates, methylation call percentages and cell type compositions across four samples, each sequencing platform is functionally equivalent from a data quality standpoint. Integration of all cells sequenced from this preparation from both platforms to yielded generate a

As a note to the reviewer, all of our current studies that leverage sciMETv3 are being carried out using Ultima Genomics due to the comparable results and substantially cheaper sequencing costs. This was decided based on the methylation calling and cluster composition, which is comparable between the platforms and the metric on which we focus our analyses.

2. You likely needed to make the libraries compatible with Ultima. Usually, the primers that are used to convert the libraries (at least in case of 3'-scRNA-seq libraries) bind the R2 region prior to the index sequence. Was that the case here as well? If yes, was a new index added to individual libraries and demultiplexing was done using those newly added indices?

The reviewer brings up an important clarification regarding how the libraries were prepared for Ultima vs the standard Illumina workflow. In our case, we never generate full Illumina libraries and convert to Ultima, but leverage the PCR to append Ultima-specific primers. Our Ultima libraries never reach a stage where they can be sequenced on an Illumina instrument.

We now state this more clearly in the text and have modified the molecular diagram portion detailing the final PCR and how it differs between the two platforms. Text Modification:

future processing (Fig. 3A). One plate was carried through bisulfite conversion, adapter ligation and PCR using primers established in previous experiments that append Illumina sequencing primers. The second plate was processed for Ultima-based sequencing using bisulfite conversion for 88 wells, and 8 wells carried through enzymatic conversion. Adapter ligation was then performed followed by PCR using primers that append sequencing primers specific to the Ultima sequencing platform. Importantly, the Ultima-based PCR (Fig. 1B, bottom) is direct Ultima library preparation and not a platform-conversion approach. Beyond alternate primer sequences, the other major design difference was to append the PCR index on the same side of the molecule as the tagmentation and ligation indexes so that the single-end reads produced by Ultima sequencing will read through all three indexes prior to the genomic DNA insert, maximizing the number of reads that will contain all three index sequences.

Figure 1B Modification:

3. Do you notice any bias towards any particular allele using sciMETv3? Or, can you potentially analyze the data at the allelic level?

Allele analysis of bisulfite-converted data is challenging due to the conversion process; however, genomic loci and variants that do not include cytosines (ie – A<->T variants), can be evaluated directly. We assessed variants in this category, as well as imposing stringent coverage and quality filters and observed a mean non-reference allele frequency of $50.01 \pm 0.04 \%$, suggesting minimal bias. This is consistent with the absence of strand bias in coverage (49.6% top strand). We are not able to assess our accuracy on these variant calls, as the individuals profiled have not been genome sequenced, which would be required for any robust assessment of allelic methylation status. We now comment on these statistics in the relevant Results sections:

mean coverage of 2.73 million total cytosines covered per cell and minimal strand bias (49.6% of reads aligned to top strand). Of these, 269 were human and 24 were mouse, with zero cells identified as doublets,

And:

dimensionality reduction and clustering in brain single-cell DNA methylation datasets^{7,11} (Figs. 3C, S1). The additional coverage in these datasets compared to the initial pilot studies allowed us to assess allelic-bias that may be present in the sciMETv3 assay. We evaluated called variant positions that are not impacted by bisulfite conversion (A to T and T to A transversions) and observed a mean non-reference allele frequency of \$50.01 \pm 0.04 \%\$, suggesting minimal allele bias from the assay or alignment.

Allele-specific analysis is particularly challenging with the sciMETv3 technology, largely due to the generally short library fragment sizes, limiting the proportion of reads that contain detectable heterozygous variants as well as informative methylation loci. We now detail this in the Discussion section as a noted limitation of the technology.

with sodium bisulfite. This longer fragment size may make allele-specific analysis of methylation more viable, as the short fragment length provided by the standard sciMETv3 workflow severely limits its utility for such studies. Resulting libraries produced comparable methylation profiles and did not exhibit any bias in

Reviewer #2:

DNA methylation is an important epigenetic modification known to control cell-type specific gene expression but is challenging to study in single cells. This manuscript addresses this challenge by presenting several modifications to a pre-existing technique. The updated method, called sciMETv3, incorporates additional index sequences into the library preparation strategy to increase the number of cells that can be studied in a single experiment (theoretically up to 10 million). The authors also show that sciMETv3 is compatible with capture-based enrichment methods, two sequencing platforms and parallel analysis of chromatin accessibility. Together, these modifications make sciMETv3 suitable for implementation in cell atlas projects.

The combinatorial indexing approach used in sciMET has several advantages over similar techniques already used in atlas-scale studies. For example, library preparation is performed on pools of nuclei as opposed to individual cells, eliminating the need for sophisticated robotic platforms. This manuscript would be strengthened by a more comprehensive comparison between sciMETv3 and alternative methods, highlighting the relative strengths and weaknesses of the different approaches.

The manuscript and figures are prepared to a very high standard and the text is succinct and easy to read. However, the extent to which sciMETv3 extends on past work from this group and others has been exaggerated by omitting key references, and by limiting the discussion to the strengths of sciMETv3.

We thank the reviewer for the positive comments and appreciate the critique regarding past atlas-scale work, primarily from the Ecker lab. Those studies inspired the sciEMTV3 technology and are incredibly valuable resources to the community. Notably, those studies require thousands of 384-well plates and extensive robotics to achieve, which we now contrast with sciMETv3 while still highlighting the incredible value of the prior studies. We detail these changes below in the specific comments that were provided.

Specific Comments:

1. The authors present sciMETv3 as compatible with atlas-scale studies of DNA methylation, but do not pay due credence to alternative methods used in published atlas-scale studies.

a. Line 27-28: The authors claim that "robust and cost-effective techniques to produce atlas-scale datasets have not been realized for DNA methylation", but several atlas-scale datasets have already been produced (e.g. Liu et al Nature 2021 & Liu et al Nature 2023 624:336). Please rephrase this section of the text to acknowledge that atlas-scale DNA methylation datasets are already available.

We thank the reviewer for pointing this out. Indeed, prior atlas work from the Ecker Lab is robust and atlas-scale, yet largely unachievable by all but the largest groups with extensive automation capabilities and budgets. We have modified the motivation section to the following:

Motivation

DNA methylation forms a basal layer of epigenomic regulatory control, shaping the genomic permissiveness of mammalian cells during lineage specification and development. Aberrant DNA methylation has been associated with myriad health conditions ranging from developmental disorders to cancer. The high cell type specificity necessitates analysis at the single-cell level, much like transcription or other epigenomic properties. ~~While atlas-scale single-cell DNA methylation datasets have been achieved as a part of the BRAIN Initiative, this work required extensive time, automation and resources, making such studies impossible to the vast majority of research programs. However, robust and cost-effective techniques to produce atlas-scale datasets have not been realized for DNA methylation.~~ Here, we directly meet this need by introducing sciMETv3, a high-throughput protocol capable of producing hundreds of thousands of single-cell DNA methylation profiles in a single experiment.

We also modified the Summary text:

Single-cell methods to assess DNA methylation have not yet achieved the same level of cell throughput per-experiment compared to other modalities, with large-scale datasets requiring extensive automation, time and other resources. Here, we describe sciMETv3, a combinatorial indexing-based

As well as the Introduction text:

methyated cytosines are protected from this process. The complexity of conversion protocols makes single-cell approaches particularly challenging, with most methods requiring the deposition and processing of individual cells into their own reaction compartments for conversion and then initial processing steps¹⁻⁵. Large-scale efforts, such as those carried out by the NIH BRAIN Initiative have achieved atlas-scale datasets^{8,10,15,27}, however, these studies required extensive automation, time and financial resources to process the thousands of microwell plates and column cleanups, all of which are well beyond what a typical lab could accomplish. The profound insight into neuronal DNA methylation biology and value as reference atlas of these efforts motivate the need for a technology that can be deployed by diverse research programs and enable the extension of such insight into other areas of biology. We previously developed techniques to

b. In the Discussion, the authors should compare the strengths and weaknesses of their method to those of alternative methods.

We have modified the Discussion text to lead with the importance of the prior atlas datasets that motivated sciMETv3 and include comparison information:

In recent years large-scale DNA methylation atlases have enabled valuable insight into the contributions of DNA methylation to the regulatory landscape^{8,10,15,27}. These studies were achieved using extensive automation and other resources well beyond the means of a typical research program. Here, we describe sciMETv3, a robust technology for the production of atlas-scale single-cell DNA methylation datasets capable of delivering library sizes in the 100's of thousands of cells. The primary advantage of sciMETv3 over technologies leveraged to produce previous atlas-scale datasets is that a ready-to-sequence library containing comparable cell throughput (e.g. 500,000 cells) can be produced by a single individual over a few days without any special equipment and utilizing a total of seven 96-well plates plus a single 96-well column cleanup. In contrast, prior atlas technologies and similar methods that require a single well per cell, requires a minimum of 1,302 or 5,208 microwell plates (384-well or 96-well, respectively) to be processed at each step until pooling can be carried out, likely taking several months and substantial other resources to achieve the same scale. Notably, the prior atlas-scale studies, as part of the NIH BRAIN Initiative, sequenced each library to a greater depth per cell than we demonstrate here; however, we achieved our desired cell type clustering granularity at the depth produced, though additional sequencing could be carried out to increase depth to a level approaching that of the BRAIN Initiative studies. Further gains in coverage could also be achieved by leveraging our previously-described sciMETv2 variant that leverages linear amplification (sciMETv2.LA) to append the reverse adapter which can provide as much as a 10-fold library complexity increase, yet at the cost of a longer protocol⁷. Such approaches are only relevant when very high per-cell coverage is desired at which point the primary factor lies in sequencing cost.

Beyond the cell throughput advances, ~~We~~ we demonstrate that sciMETv3 is compatible with capture-

2. Capture-based enrichment is used to reduce the sequencing demand of single-cell DNA methylation analysis, but limited detail is provided by the authors.

We appreciate the reviewer's suggestions to add clarity and detail to the capture work.

a. The authors list "Compatibility of sciMETv3 with capture techniques to reduce sequencing burden" as a highlight of this paper. This gives the impression that this is the first study to combine single-cell DNA methylation analysis with capture-based enrichment methods. However, the authors have recently published sciMET-cap in a July 2024 Genome Biology paper. The authors should revise this manuscript to appropriately describe past work and to highlight the novel contributions of this paper.

We did not intend for this work to imply it was the first to achieve capture workflows with sciMET and thank the reviewer for pointing out that it could be mistaken as such. We now specifically state the original "sciMET-cap" technique in the highlight bullet points and have modified the Results text accordingly:

We next assessed the full workflow and platform versatility of sciMETv3 by carrying out a preparation on a human brain specimen (cortex, BA 46) with the goal of expanding into enzymatic-based conversion as well as leveraging our previously-described sciMET-cap technology that enriches for a roughly 125 Mbp set of regulatory loci in order to enrich for higher variable regions and subsequently reduce the sequencing depth required to achieve comparable cluster granularity⁹. We leveraged 96 tagmentation and ligation indexes and

As well as the diagram in Figure 1C, which now includes the prior work's citation:

b. Line 278: The authors explain that capture-based techniques can reduce the sequencing demand of DNA methylation atlas studies, but it is not clear if this is unique to sciMETv3. Are other (plate-based) methods for single-cell analysis of DNA methylation also compatible with capture-based techniques?

This is a good point, it is quite likely that such capture methods would be compatible with plate-based single-cell methylation workflows; however we have not attempted such experiments and they would likely require method-specific optimization which we believe is outside of the scope for this paper.

c. Line 278: By how much is the sequencing demand reduced if one uses capture-based techniques?

This is a point that we cover extensively in the original sciMET-cap publication, which we now reference more frequently in this manuscript according to the reviewer's suggestions. We found that the fold reduction in sequencing required ends up tracking very similar with the fold enrichment achieved; however, this is when assessing CG methylation. For brain, where CH methylation is abundant, the genome-wide nature of it does not benefit from capture and therefore the gains are reduced. We now address this in the Discussion section:

Beyond the cell throughput advances, We demonstrate that sciMETv3 is compatible with capture-based techniques which allow for a reduced amount of sequencing to produce robust cell type clustering, tracking in fold-reduction with the fold-enrichment achieved by the capture which was demonstrated for both human brain and PBMC tissues when focusing on CG methylation. In brain, where CH methylation is abundant and genome-wide, as opposed to the enrichment at regulatory loci observed with the CG context, the gains from capture are less pronounced, reducing its utility. Our assessment of sciMETv3 with capture allowed for approximately 86,000 single-cell libraries to be multiplexed within a single capture reaction without a substantial reduction in on-target capture rate, achieving a 6.2-fold enrichment. Notably, the capture workflows produce sufficient off-target coverage to provide genome-wide methylation calls when cells are aggregated at the cluster level, mitigating the limitation of capture techniques where non-targeted regions are missed.

d. Line 281: What is the off-target coverage associated with the capture-based techniques?

We achieve a slightly lower, yet comparable on-target rate as our prior sciMET-cap technology. We now include this in the Results section text:

stage. Eight wells were taken through bisulfite conversion, reverse adapter ligation and PCR. All eight wells (estimated cell n = 6,000) were taken through the capture workflow followed by sequencing, producing 5,805 QC-passing single-cell DNA methylation profiles with a comparable target fold enrichment to sciMET-cap (6.2-fold versus 7 to 10-fold⁹), which translates to a similar rate of on-target reads at 27.7% versus 30-45%.

3. The authors present data to demonstrate that sciMETv3 is compatible with parallel analysis of chromatin accessibility by ATAC-seq; however, the resulting data is of reduced quality for both modalities. The authors acknowledge that the sciMETv3 data had "elevated noise" (line 257) and that the "transcription start site enrichment (TSSe) was relatively low" (line 235) in the ATAC-seq data.

We did not want to oversell the performance and believe it is important to highlight the shortcomings of the co-assay while also demonstrating the utility. We now expand further on the 'noise' and emphasize the role for which we expect the co-assay will be most useful in the Discussion: serving as a bridging modality. Additional details in comments below.

A possible alternative to this approach would be a Nucleosome Occupancy and Methylation (NOME) sequencing approach in which a bacterial GpC methyltransferase is used to mark accessible chromatin prior to library preparation. Have the authors considered combining sciMETv3 with a NOME-based approach to assess DNA methylation and chromatin accessibility in the same single cells?

The reviewer brings up an excellent point and one we had thought about previously, including carrying out several sciMETv2 preparations that incorporated the NOME component. The primary disadvantage of the NOME technique is that there is no enrichment for open loci, meaning that to assess the ~1-5% of the genome that falls within open chromatin loci, a random read from the genome must hit the locus. When libraries are sequenced to very high depth, a substantial number of loci may be covered; however, at a lower depth per cell very few will be assessed, limiting any analysis to the pseudo-bulked cluster level. In contrast, the incorporation of ATAC specifically enriches for open chromatin reads, increasing the number assayed per-cell and enabling chromatin accessibility to be leveraged at the single-cell level. One disadvantage of ATAC is that there is no information observed at closed regions, whereas the absence of methylation at GpCH sites at a locus indicates inaccessibility. We now discuss NOME methods in both the Results and Discussion section, as well as comment on the possibility to combine both, taking advantage of the inaccessibility readings from the NOME readout at the cluster-level along with the enrichment advantages of ATAC.

Results paragraph added to the start of the section:

Chromatin accessibility has been proven to be valuable property for assessing the regulatory landscape of cells^{13,19,22,24,25}. Advances in technology platforms to assess chromatin accessibility using transposase-based workflows (ATAC) at the single-cell level have enabled large-scale atlases to be produced across multiple tissue types, providing a valuable reference resource to aid in cell type assignment via marker gene assessment or direct integration. Previously, technologies have been developed to assay nucleosome occupancy (similar to chromatin accessibility) alongside DNA methylation (NOMe) by pre-treating nuclei using a bacterial DNA methyltransferase to artificially methylate cytosines in the GC context at both accessible chromatin and histone linker regions²⁰. Assessing methylation levels at these sites, excluding those in the GCG context, allows the ability to determine if a site was accessible by the methyltransferase. These technologies have been extended to the single-cell level^{27,29,31}; however, the lack of enrichment for regulatory loci, as is the case for ATAC-based technologies, make coverage of the 1-5% of the accessible genome²⁴ extremely sparse on a per-cell basis, relegating any chromatin analysis to the cell type or cluster level³³.

And in the Discussion:

combination of both to be used as genome-wide DNA methylation. Unlike NOMe-based techniques that encode accessibility using artificial GpC methylation, our strategy enriches for accessible loci using ATAC-based workflows, providing enough information per cell to enable cell-level ATAC analysis as opposed to cluster-level. Additionally, the technique is applicable to brain and embryonic stem cells where GpC methylation is common due to the presence of noncanonical CH-context methylation genome-wide^{42,43}, complicating NOMe-based approaches. Overall, the data quality of scMET+ATAC is lower for each modality

4. Line 206-209: The authors explain that 12.1% and 28.6% of HMRs overlap promoters and enhancers, respectively. Is this overlap more or less than expected based on the number and genomic locations of promoters and enhancers? In other words, are HMRs enriched in enhancers?

Indeed, HMRs are significantly enriched at both enhancers (1.74-fold) and promoters (8.07-fold). We also expanded this analysis to ENCODE DHS sites, for which 68.6% of HMRs overlapped with an enrichment of 1.38-fold. We add this information to the Results text along with supporting citations on prior observations of this enrichment and differences in cell type variability of enhancers and promoters.

To characterize these distinct patterns, we assessed cell type clusters (n = 31) genome-wide for hypomethylated regions (HMRs; methods). In total, 155,110 distinct HMRs were identified with 65,161 (42.0%) unique to a single cluster. A total of 106,424 (68.6%) overlapped ENCODE DNaseI Hypersensitivity sites²⁴ (1.38-fold genomic enrichment, $p < 2.2e^{-10}$, Hypergeom), consistent with the majority of these sites serving a regulatory role. Of these, 18,800 (12.1%) HMRs overlapped promoter regions (8.07-fold genomic enrichment, $p < 2.2e^{-10}$) with only 1,463 (7.8% of promoter HMRs) unique to a cluster and a mean of 17.5 clusters exhibiting hypomethylation at HMRs, indicating a propensity for cross-cell type promoter hypomethylation, regardless of expression status. In contrast, of the 44,304 (28.6%) of enhancer-overlapping HMRs (1.74-fold genomic enrichment, $p < 2.2e^{-10}$), 14,668 (33.1% of enhancer HMRs) were cell type specific and a mean of 4.7 cell types exhibited hypomethylation at these HMRs, suggesting increased cell type specificity versus promoter elements, consistent with previous enhancer characterization studies¹⁰ (Fig. 4G).

5. Lines 271-274: The authors describe DNA methylation variance at promoters and enhancers defined using ATAC-seq data from the matched cell population. The results show that enhancers with intermediate levels of DNA methylation have the highest variance. This finding is consistent with several previous reports, which should be cited here.

This is certainly true. We now conclude the results paragraph stating that our observations are consistent with prior literature.

Paired ATAC and genome-wide DNA methylation enables the assessment of both open and closed chromatin for DNA methylation status, as opposed to methods that conduct bisulfite conversion only on ATAC-derived reads, providing insight into the regulatory status of loci across all cell types and not just those that exhibit open chromatin. To assess these interactions, we leveraged the methylation-based cell typing to produce aggregated ATAC tracks, producing distinct cell type-specific accessibility patterns at marker genes (Fig. 5H). We then assessed ATAC peaks called from the data for methylation status across cell types, splitting out the ATAC peaks by promoters and enhancers (Fig. 5I). Between these categories, methylation was less variable at promoter regions, with nearly all cell types exhibiting hypomethylation. This low-variance hypomethylation population was present in the enhancer peak set, yet only for a minority of peaks, with the large majority exhibiting higher methylation variance where a majority of cell types exhibited hypermethylation. This observation is consistent with previous studies that have shown enhancers with intermediate DNA methylation have the highest tissue and developmental variance^{37,39-42}.

6. Lines 257, 309 and Figure S3D: In the combined sciMETv3/ATAC analysis, there is a cluster of cells enriched for high doublet scores. The authors refer to this observation using the somewhat ambiguous term "noise". Can the authors explain why there may be a higher doublet rate in the multi-modal analysis? Line 256-258 should be revised to describe the observation in less ambiguous terms.

We now include more detail on the source of the 'noise' which we believe is due to ruptured nuclei and possible crosstalk from ambient chromatin fragments. This is stated in the Results section:

We next processed the DNA methylation side, producing cell groupings similar to the assigned cell types from the ATAC modality (Fig. 5F). The methylation modality was combined with our previous sciMETv3 dataset produced on the same individual, which produced substantial overlap except for a single cluster that was able to be filtered out using our doublet detection model, suggesting elevated noise in the form of ruptured nuclei and ambient chromatin crosstalk in the dataset compared to the unimodal sciMETv3 workflow (Fig. S3B-D). This form of noise is likely to occur more frequently when nuclei undergo two rounds of fragmentation versus a single fragmentation in either unimodal assay. We then leveraged the cluster identities

And the implications are further described in the Discussion section:

the data quality of sciMET+ATAC is lower for each modality than when performed on their own, as represented by a lower TSS enrichment value in the ATAC modality and the presence of noise in the methylation modality likely due to nuclei rupture and elevated ambient chromatin fragments. However, the

And:

content produced using the sciMET assay. Taken together, we believe that the sciMET+ATAC workflow will be a valuable for profiling a portion of cells in addition to the sciMETv3 workflow to bridge between datasets and facilitate cross-modality integration and cell type assignment as opposed to serving as a standalone dataset due to the reduced TSS enrichment and elevated ambient chromatin noise.

7. There is no information regarding ethics approvals in the Methods section. Was this study conducted with appropriate ethical oversight?

We now include the section below:

Ethical Approval

All tissue specimens were obtained from the NIH NeuroBioBank as a part of the NIH BRAIN Initiative Cell Atlas Network (BICAN) collections which are consented for open data release. Ethical oversight was carried out by the OHSU Institutional Review Board.

Minor Corrections, Suggestions and Queries

1. Line 61: Consider replacing unfamiliar word "amortizes". We changed the term to "distributes"
2. Figure 1E: It is confusing to have a legend symbol for "Mix" when there are no cells in this category. Consider removing this symbol from the graph. Done.
3. Line 428, 447: Use subscript for "2" in "MgCl₂" and "dH₂O". Done.
4. Line 429: "Nichols et al. 2022" is not in the reference list. Included.
5. Line 430, 442 and throughout: Degrees Celsius should be written as "°C". Done.
6. Line 444: "in situ" should be italicized Done.
7. Line 449: Please specify if the ligation barcodes are included in a kit. They are not, we now reference IDT and Supplementary File 1 for the sequences.
8. Line 497: Please check: "95°C hold / 95°C 5' / 60°C hold (lid 85°C). 20 µl of 65°C." What does 20 µl refer to? Is there an extra full stop? We reworked the text to improve clarity.
9. Line 519: Please add a citation for Trim Galore. There was no publication for TrimGalore, but we now cite the Zenodo doi for the version release.
10. Line 544: Please remove mid-word space in "using amethyst visualization functions". Done.

Referees' report, second round of review

Reviewer 1

I appreciate the authors' effort and due diligence in this well-written manuscript, and also in their responses to the reviewers' comments. My concerns have been satisfactorily addressed. Thanks!

Reviewer 2

In the revised manuscript, the authors have responded to all reviewer comments and present a more balanced assessment of the sciMETv3 method and alternative approaches.

Further revisions are not required, but I would like to offer one final comment.

I agree with the authors that the plate-based methods used in the Ecker lab are beyond the reach of most groups. However, I am not confident that the complex combinatorial-indexing approach used in sciMETv3 will be much more

accessible. I suspect that single-cell analysis of DNA methylation will remain a 'niche' technology performed by specialist groups within international consortia. Unlike single-cell transcriptomics, the biological meaning of DNA methylation can be challenging to interpret, and it will be difficult to justify the additional cost in medical research laboratories where resources are limited. I would be delighted to be proven wrong in this matter. Heather Lee.

Authors' response to the second round of review

N/A